# MQuAKE-Remastered:
# Multi-Hop Knowledge Editing Can Only Be Advanced With Reliable Evaluations

**Shaochen (Henry) Zhong**[*][♣], **Yifan Lu**[*][♣], **Lize Shao** [♣], **Bhargav Bhushanam** [∞], **Xiaocong Du** [∞],
**Louis Feng** [∞], **Yixin Wan** [†], **Yiwei Wang** [†], **Daochen Zha** [♣], **Yucheng Shi** [◇], **Ninghao Liu** [◇],
**Kaixiong Zhou** [♡], **Shuai Xu** [♠], **Vipin Chaudhary** [♠], and **Xia Hu** [♣]

[♣] Department of Computer Science, Rice University
[◇] School of Computing, University of Georgia
[♡] Department of Electrical and Computer Engineering, North Carolina State University
[♠] Department of Computer and Data Sciences, Case Western Reserve University
[†] Department of Computer Science, University of California, Los Angeles
[∞] Meta Platforms, Inc.

## Abstract

Large language models (LLMs) can give out erroneous answers to factually rooted questions either as a result of undesired training outcomes or simply because the world has moved on after a certain knowledge cutoff date. Under such scenarios, *knowledge editing* often comes to the rescue by delivering efficient patches for such erroneous answers without significantly altering the rests, where many editing methods have seen reasonable success when the editing targets are simple and direct (e.g., *"what club does Lionel Messi currently play for?"*). However, knowledge fragments like this are often deeply intertwined in the real world, making effectively propagating the editing effect to non-directly related questions a practical challenge (to entertain an extreme example: *"What car did the wife of the owner of the club that Messi currently plays for used to get to school in the 80s?"*). Prior arts have coined this task as *multi-hop knowledge editing* with the most popular dataset being MQUAKE, serving as the sole evaluation benchmark for many later proposed editing methods due to the expensive nature of making knowledge editing datasets at scale. In this work, we reveal that **up to 33% or 76% of MQUAKE's questions and ground truth labels are, in fact, corrupted in various fashions due to some unintentional clerical or procedural oversights**. Our work provides a detailed audit of MQUAKE's error pattern and a comprehensive fix without sacrificing its dataset capacity. Additionally, we benchmarked almost all proposed MQUAKE-evaluated editing methods on our post-fix dataset, MQUAKE-REMASTERED. It is our observation that many methods try to overfit the original MQUAKE by exploiting some data-specific properties of MQUAKE. We provide a guideline on how to faithfully approach such datasets and show that a simple, minimally invasive approach can bring excellent editing performance without such exploitation. Please refer to https://github.com/henryzhongsc/MQuAKE-Remastered and supplemental material for assets.

---

* Equal contribution. Work corresponds to Shaochen (Henry) Zhong <shaochen.zhong@rice.edu>.

Submitted to the 38th Conference on Neural Information Processing Systems (NeurIPS 2024) Track on Datasets and Benchmarks. Do not distribute.

# 1 Introduction

Given the widespread public-facing popularity of various Large Language Model-powered (LLM) products [Zhao et al., 2023, Yang et al., 2024], even an occasional user has likely experienced LLMs giving out erroneous answers to factually rooted, knowledge-intensive questions. While the reasons why LLMs would hallucinate such kind of misinformation is complex and still an open problem — noisy training data, model bias, out-of-distribution questions, or even simply because the world has moved on after a certain knowledge cutoff date, all likely contributed their fair share to this rather undesired character of LLMs [Huang et al., 2023, Zhang et al., 2023]— **under a practical context, *knowledge editing* is often considered the go-to remedy by delivering efficient patches for such erroneous answers** without significantly altering the LLM's output on unrelated queries [Sinitsin et al., 2020, Mitchell et al., 2022].

With the growing need to have more credible and trustworthy LLMs, a vast amount of LLM-specific knowledge editing methods have been proposed, and many of them have seen reasonable success in addressing editing targets that are simple and direct. For example, most modern knowledge editing methods can reliably edit the answer of *"What club does Lionel Messi currently play for?"* from *"Paris Saint-Germain"* to *"Inter Miami CF"* and therefore correctly reflecting the occupation status of Messi [Zhong et al., 2023].

## 1.1 Multi-hop knowledge editing poses practical significance and non-trial challenges.

However, due to the intertwined nature of different knowledge fragments, a small change in one knowledge fragment can produce ripple-like effects on a vast amount of related questions [Zhong et al., 2023, Cohen et al., 2023]. It is often a non-trivial challenge to efficiently propagate the editing effect to non-directly related questions with proper precision and locality. E.g., for a — in this case intensionally extreme — question like *"What car did the wife of the owner of the club that Messi currently plays for used to get to school in the 80s?"* Many knowledge-edited LLMs can still struggle while being fully aware of Messi's abovementioned club transfer [Zhong et al., 2023].

Prior arts have realized the practical significance of being able to edit such complex/non-direct questions upon a certain knowledge update, as different knowledge fragments are almost always deeply entangled with each other in the real world [Zhong et al., 2023, Cohen et al., 2023, Wei et al., 2024]. Meanwhile, exhausting all potential combinations of questions related to one or a few updated knowledge fragments is impractical, if not totally impossible: imagining editing an LLM for every possible question influenced by the abovementioned club transfer of Messi. Even if it is feasible, this poses high operational costs and comes with the intrinsic risks of editing a mass amount of targets; not to mention a repeated effort would be required should Messi ever opt to transfer again.

It is intuitive that a practical knowledge editing method should be able to produce correct answers to relevant factual questions with only a few updated knowledge fragments available. This task has been coined as *multi-hop knowledge editing* **with the founding, largest, as well as the most popular dataset to date being MQuAKE by Zhong et al. [2023]; serving as the sole evaluation backbone for many proposed modern editing methods** due to the expensive nature of making counterfactual and temporal datasets at such a scale ($> 10,000$ cases provided, more about the dataset statistics in Table 6). Note that such expansiveness is further multiplied given the abovementioned ripple effect of multi-hop question answering, as one knowledge update of a subquestion can potentially lead to multiple updated answers across a large number of cases.

## 1.2 Unfortunately, MQuAKE is flawed due to unintentional clerical and procedural errors — we fixed/remade it and re-benchmarked almost all proposed multi-hop knowledge editing methods.

While MQuAKE is the founding dataset of multi-hop knowledge editing tasks and very much brings life to this vital subject, through a comprehensive audit, we reveal that **up to 33% or 76% of MQuAKE questions and ground truth labels are, in fact, corrupted in various fashions due to some unintentional clerical or procedural errors**; which inevitably cast doubts on the effectiveness of developed methods (especially the ones that solely) evaluated on MQuAKE, and **present as a**

**hidden peril to the field's progress as such flaws are largely unknown to the knowledge editing community before our work.** We highlight that the flaws of MQUAKE is an already massive yet constantly growing issue, as MQUAKE is one of the fastest-growing datasets in terms of adaptation in the editing community, yet, the task it is trying to tackle — building more reliable LLM — is without a doubt crucial aspect of NLP development. To pave the way for future advancement of multi-hop knowledge editing, we present our work with the following contributions:

- **A comprehensive audit of MQUAKE:** We are the first to present a comprehensive audit of the existing errors within MQUAKE [Zhong et al., 2023], bringing awareness to the knowledge editing community regarding this popular dataset with significant task importance attached.
- **Fix/remake MQUAKE to MQUAKE-Remastered:** We present the only available fix/remake that not only patches all discovered errors, and done so without sacrificing the intended intensity and capacity of the original MQUAKE whenever possible.
- **Extensively re-benchmark of almost all existing multi-hop knowledge editing methods:** Given the currently existing reports based upon the original MQUAKE are flawed reflections of such proposed methods' capability, we additionally re-benchmark almost all existing multi-hop knowledge editing methods that are available against our MQUAKE-REMASTERED datasets.
- **Guidance for future multi-hop knowledge editing development.** Upon our extensive re-benchmark results, we observe that many proposed multi-hop knowledge editing methods intentionally or unintentionally overfit the original MQUAKE dataset by applying data-specific operations that are largely unique to the MQUAKE dataset family. We provide guidance on how to faithfully approach these datasets and additionally show that a simple, minimally invasive approach with no such operations can also achieve excellent editing performance.

## 2 Preliminary

### 2.1 Background of MQUAKE

MQUAKE (Multi-hop Question Answering for Knowledge Editing) is a knowledge editing dataset focusing on the abovementioned multi-hop question answering tasks proposed in Zhong et al. [2023], where every case of MQUAKE is a multi-hop question made by a chain of single-hop subquestions. Specifically, MQUAKE is constructed based on the Wikidata:RDF dataset [Vrandečić and Krötzsch, 2014], which, in its rawest format, is a knowledge graph consisting 15+ trillion of Resource Description Framework (RDF) triples[1]. MQUAKE essentially builds a much more concise subgraph with only 37 manually elected common relations and top 20% of the most common entities, where a walk of $\{2, 3, 4\}$-hop on this subgraph can form a case (which is a chain of $\{2, 3, 4\}$ single-hop subquestions connected together) in the MQUAKE dataset.

MQUAKE is presented as two (but in practice, it is essentially three) sub-datasets: MQUAKE-CF and MQUAKE-T. The former focuses on counterfactual tasks, while the latter on temporal changes. We highlight that there is also a MQUAKE-CF-3K dataset, which is a subset of MQUAKE-CF that only contains 3,000 cases in total (with 1,000 cases for $\{2, 3, 4\}$-hop questions respectively). Authors of MQUAKE evaluate their proposed method, MeLLo [Zhong et al., 2023], upon this MQUAKE-CF-3K dataset, citing limited compute resources; which then become an unspoken standard practice for the majority of the later proposed multi-hop knowledge editing methods [Gu et al., 2024, Shi et al., 2024, Wang et al., 2024, Anonymous, 2024, Cheng et al., 2024]. Due to the very popularity of this sub-sampled dataset, we provide our error analysis mostly based on MQUAKE-CF-3K and MQUAKE-T in the following §3. For interested readers, we additionally provide the same error analysis upon the full MQUAKE-CF in the Appendix B.2, which is only more drastic than MQUAKE-CF-3K due to MQUAKE-CF being a much larger superset of the already compromised MQUAKE-CF-3K. We also collect the dataset statistics in Table 6 to provide a numerical overview of the composition of all three MQUAKE datasets.

---

[1] https://www.wikidata.org/wiki/Property:P10209

## 2.2 Evaluating using MQUAKE

Datasets like MQUAKE-CF or MQUAKE-CF-3K are often evaluated against different "editing intensity," which is controlled by how many cases among all tested cases are considered "edited," mimicking different levels of deviation between the learned knowledge stored in the LLM and the desire edited knowledge. This is a sound practice because proper knowledge editing methods should perform well when different numbers of knowledge fragments are edited, as it is equally important to navigate when a significant amount of knowledge is updated, as well as to recognize the few edited knowledge and limit their influence from unrelated unedited knowledge with proper editing locality.

In its original paper, MQUAKE-CF-3K is evaluated when $\{1, 100, 1000, 3000\}$ of its 3,000 cases are edited, similarly, MQUAKE-T is evaluated when $\{1, 100, 500, 1868\}$ of its 1,868 cases being edited, forming an experiment report like Table 5. This kind of report granularity (a gradual coverage from a few edits to all cases being edited) is also adopted by the majority of later proposed multi-hop knowledge editing methods, either in full [Anonymous, 2024] or in spirit with different subsample settings [Gu et al., 2024, Wang et al., 2024, Shi et al., 2024, Cheng et al., 2024, Mengqi et al., 2024]. In this work, we report at an even finer level of granularity for maximum cross-reference potentials.

# 3 Auditing MQUAKE

In this section, we present a comprehensive audit of the error pattern that existed in MQUAKE-CF-3K and MQUAKE-T [Zhong et al., 2023]. We specifically note that our audit is there to provide a better understanding to the knowledge editing community, especially when digesting methods evaluated on these datasets. **Our audit is not to discredit the contribution of MQUAKE, or any of the proposed methods evaluated on MQUAKE.** We recognize the fact that no dataset can be perfect, especially when it is intrinsically hard to collect large-scale counterfactual and temporal datasets.

## 3.1 Intra Contamination between Edited Cases and Unedited Cases

As discussed in §2.2, having a gradual evaluation coverage from a few to all cases being edited like Table 5 makes sense for as an evaluation granularity. However, one critical issue is that $k \in \{1, 100, 1000, 3000\}$-edited cases (supposed MQUAKE-CF-3K) are randomly sub-sampled from the 3,000 total cases. Thus, **there is no guarantee that the $k$-edited cases and $(3000 - k)$ unedited cases would require two disjoint sets of knowledge and, therefore, risk contamination.**

For a concrete example, consider the following two multi-hop questions from MQUAKE-CF-3K (we also additionally provide the subquestion breakdown and intermediate answers of the two questions for better presentation, we note that such auxiliary information is not part of the instruction visible to the question-answering LLM):

- `case_id:245` (unedited): *What is the official language of the country where Karl Alvarez holds citizenship?*
    - ◇ What is the country of citizenship of Karl Alvarez? USA.
    - ◇ What is the official language of United States of America? American English.

- `case_id:323` (unedited): *What language is the official language of the country where Wendell Pierce holds citizenship?*
    - ◇ What is the country of citizenship of Wendell Pierce? USA.
    - ◇ What is the official language of United States of America? American English.

For both questions, the correct pre-edited answer should be *"American English."* As both Karl Alvarez and Wendell Pierce are US citizens, and the official language of the US is American English. However, suppose `case_id:323` is sampled as an edited case while `case_id:245` remains unedited, we will be provided with the additional triple containing the knowledge of *"The official language of United States of America is Arabic."*

Since the unedited `case_id:245` and the edited `case_id:323` share the same subquestion of *"What is the official language of United States of America?"* The answer of `case_id:323` will be rightfully updated to *"Arabic"* per the new knowledge. However, the unedited `case_id:245` still considers the

original answer *"American English"* to be correct, and is therefore contaminated by the edited case `case_id:323` in an unintended fashion. This is problematic because a successful knowledge editing method should be able to retrieve the edited knowledge — *"The official language of United States of America is Arabic"* — upon the relevant questions (in this case the shared one), and thus answering *"Arabic"* to `case_id:245`. This is technically correct, but in conflict with MQUAKE-CF-3K's label, causing inaccurate experiment readings.

**We further note the above-illustrated contamination is not a cherry-picked fluke, but rather a wild-spread error.** Here, we sample $\{1, 100, 1000, 2000, 3000\}$-editing targets from MQUAKE-CF-3K using random seed 100, and find the following error statistics in Table 1.

Table 1: Error statistics of MQUAKE-CF-3K and MQUAKE-T [Zhong et al., 2023] in terms edited cases contaminating unedited cases. $k$-edited means $k$ cases out of the total dataset are edited.

| # of Contaminated | MQUAKE-CF-3K | | | | | MQUAKE-T | | | |
|---|---|---|---|---|---|---|---|---|---|
| | 1-edit | 100-edit | 1000-edit | 2000-edit | 3000-edit | 1-edit | 100-edit | 500-edit | 1868-edit |
| Cases | 0 | 2,013 | 1,772 | 910 | 0 | 29 | 1421 | 1327 | 0 |
| Subquestions | 0 | 2,706 | 3,075 | 1,664 | 0 | 29 | 1421 | 1327 | 0 |

It is observable from Table 1 that **even a small number of edited cases will cause a concerningly large contamination to unedited cases and subquestions, where 67% and 76% of all cases from MQUAKE-CF-3K and MQUAKE-T are contaminated with just 100 cases being edited**, introducing a significant distortion to the reported experiment results.[2]

We additionally note while this edited-to-unedited intra-contamination is reducing with $k$-edit growing, this does not imply a diminishing of issue, but rather a simple by-product of a larger $k$ implies a lesser $(3000 - k)$, leaving fewer unedited cases as potential contamination victims. In the extreme case of 3000-edit, there is 0 edited-to-unedited contamination because there is no unedited case left in MQUAKE-CF-3K to be the victim. But 3000-edit has the most edited-to-edited inner contamination, more on this in the following §3.2.

## 3.2 Inner Contamination between Different Edited Cases

Other than edited cases contaminating unedited cases (§3.1), contamination might also happen among multiple edited cases because a certain subquestion presented in different edited cases can be edited in some but unedited in others[3]. For brevity, we leave the example walkthrough in Appendix B.1.

Table 2: Error statistics of MQUAKE-CF-3K [Zhong et al., 2023] in terms edited cases contaminating each others. $k$-edited means $k$ cases out of the total 3,000 cases are edited.

| # of Contaminated | 1-edit | 100-edit | 1000-edit | 2000-edit | 3000-edit |
|---|---|---|---|---|---|
| Cases | 0 | 14 | 265 | 619 | 998 |
| Subquestions | 0 | 14 | 337 | 854 | 1,399 |

This type of contamination is, once again, universally visible in MQUAKE, as shown in Table 2; which is very much a flipped version of Table 1. With $k$-edit growing, there are more edited cases, thus more edited-to-edited contamination, as there are more potential victims. Notably, **under the 3000-edit tasks, almost one-third (998/3000, $\approx$33%) of the evaluated cases are contaminated**, which again introduces distortion to the reported experiment results. We omit the report on MQUAKE-T here because there is only one edit-to-edit contamination when all 1,868 cases from MQUAKE-T are edited (`case_id:424`).

---

[2]We note that in Zhong et al. [2023], "$k$-edit" means only $k$ of edited cases are evaluated, without any unedited cases. We evaluated both to better reflect the locality of different knowledge editing methods.

[3]Note, an edited case does not require all of its subquestions being edited, but merely one or more of it (Table 6)

## 3.3 Conflicting Edits

The two types of contamination introduced in §3.1 and §3.2 are indeed subtle and hard to detect, as they hide between the retrieval scope of different edited cases, which is further complicated when only a subset of cases are edited. However, MQUAKE-CF-3K also includes some straightforward conflicts, such as for the subquestion *"Which company is Ford Mustang produced by?"* we have the following edits:

◇ `case_id:2566` (edited): ~~Ford Moter Company~~ Nintendo.
◇ `case_id:231/2707` (edited): ~~Ford Moter Company~~ Fiat S.p.A.

This is going to cause a direct conflict when `case_id:2566` and any of the `case_id:231/2707` are both selected as edited cases, as they shall confuse any knowledge edited LLM for having two answers to the same questions. Fortunately, such types of errors are rather minuscule in MQUAKE-CF-3K, with the abovementioned Ford Mustang question and three cases being the only affected data samples.

## 3.4 Missing Information in Multi-hop Question Instructions

As mentioned in §2, the MQUAKE dataset is built upon a severely filtered Wikidata:RDF knowledge graph [Vrandečić and Krötzsch, 2014]. Specifically, the triples of a certain $\{2, 3, 4\}$-hop walk on this subgraph are then fed into a `gpt-3.5-turbo` model to generate the multi-hop question instruction in a natural language format; such generation are repeated for three different times in case any of the generated question instructions becomes incomprehensible. For every case evaluation, an LLM is considered right should it correctly answer against any three of the multi-hop question instructions [Zhong et al., 2023].

However, while repeating generation three times definitely reduces the chances of having incomprehensible question instructions, we noticed some of such instructions in MQUAKE are still incomplete. We take the following triple set and its generated 3-questions as an example:

• `case_id:546` (unedited): We have a 2-hop triple chain of (`Albert Mohler, employer, Southern Baptist Theological Seminary`) and (`Southern Baptist Theological Seminary, religion or worldview, Southern Baptist Convention`). MQUAKE-CF-3K provides the following generated multi-hop questions:
   ◇ Generation #1: *What religion is Albert Mohler associated with?*
   ◇ Generation #2: *Which religion does Albert Mohler follow?*
   ◇ Generation #3: *With which religious faith does Albert Mohler identify?*

It is clear that all three generated questions omit the part mentioning which company/institution Albert Mohler is employed by and essentially reduce themselves to single-hop questions, where a correct generation should read like *"What religion is Albert Mohler's employer associated with?"* Without the complete question, suppose there is an edit on Albert Mohler's employer (which there indeed is one), the final answer would likely change. However, with question instruction omitting such information, even the best knowledge-edited LLM cannot answer the question correctly with a faithful approach.

As a general analysis, we find **the natural language question instructions of 672 cases in MQUAKE-CF-3K are missing information in comparison to their raw triplet chain.** This number is counted in the sense that one or more pieces of information present in the triple chain are missing from all three variants of the generated natural language instruction. Similarly, there are 2,830 and 233 cases of erroneous instructions in MQUAKE-CF and MQUAKE-T, respectively.

## 3.5 Duplicated Cases

The last kind of error we discovered in MQUAKE is simply unintended duplication — i.e., two or more cases sharing the same start subjects, edited facts, chain of triples, and final answer. We discovered 47, 4, and 4 cases of duplication, respectively, in MQUAKE-CF, MQUAKE-CF-3K, and MQUAKE-T.

# 4 Remastering MQuAKE

In this section, we illustrate how we modified and improved the MQuAKE dataset to MQuAKE-Remastered with various fixes on the data samples themselves, as well as providing utility modules to facilitate how one interacts with such datasets.

## 4.1 Hard Corrections

Three types of error existing in MQuAKE can be fixed once and for all with some careful hard corrections, they are namely Conflicting Edits (§3.3), Missing Information in Multi-hop Question Instructions (§3.4), and Duplicated Cases (§3.5). For Conflicting Edits and Duplicated Cases, since there are only a few such errors (<50 per type per dataset), we employ some manual corrections to address these errors: in the former case, we flip the minority edits to align with the majority edits (and adjust their answers to their subsequence subquestions, should there be any); in the latter case, we simply remove such duplicated cases (except for MQuAKE-cf-3k, which we manually select 4 more cases from MQuAKE-cf to keep the dataset having 3,000 cases in total and a 1,000 cases for $\{2, 3, 4\}$-hops). For the Missing Information in Multi-hop Question Instructions errors, we rewrite such natural language question instructions and then replace the original information-missing instructions.

## 4.2 Dynamic Masking for Maximum Coverage: MQuAKE-Remastered-cf, MQuAKE-Remastered-cf-3k, and MQuAKE-Remastered-t

Due to the contamination count of Intra Edited-to-Unedited Contamination (§3.1) and Inner Edited-to-Edited Contamination (§3.2) tend to grow in the opposite direction as shown in Table 1 and 2, it is impossible to find a fix within the current MQuAKE that can address both issues without significantly decreasing the dataset size. As an alternative, we develop an API that will take a `case_id` and an `edited_flag` as input, respectfully indicating the evaluating case-in-question and whether this case is considered edited; our API shall then return a set of triples that are contamination free by dynamically masking out the conflicting edits from other cases. After such, the user may build up an editing knowledge bank upon such triplets and conduct evaluations for any memory-based knowledge editing methods without losing any of the 9,218 cases from MQuAKE-cf or 1,868 cases from MQuAKE-t.

Specifically, once `case_id`-of-interest is given, our API would loop through all of its subquestions and identify if any of such subquestions is considered edited under another case. If there is a hit, the triple with respect to such edited subquestions is then removed from the bank of edited triples. This dynamic masking mechanism would ensure all cases within the original MQuAKE be usable against memory-based knowledge editing methods. **However, the drawback of masking is it won't support parameter-based knowledge editing methods**, where weight update is required. We additionally provide a MQuAKE-Remastered-cf-6334 to address the need for such methods (Appendix C.1).

# 5 Benchmark and Discussion

Given almost all proposed multi-hop knowledge editing methods are evaluated on the original, error-contained, MQuAKE datasets. Here, we provide a re-benchmark of those methods against post-fix MQuAKE-Remastered datasets for a more reliable reporting of each method's performance.

## 5.1 Experiment Coverage

**Compared Methods** In this work, **we aim to cover most, if not all, open-sourced knowledge editing methods evaluated on the original MQuAKE.** To the best of our knowledge, this screening criteria include MeLLo [Zhong et al., 2023] and PokeMQA [Gu et al., 2024] as methods specifically proposed to target this multi-hop knowledge editing problem and evaluated on MQuAKE. We additionally include ICE [Cohen et al., 2023] and IKE [Zheng et al., 2023a] as these are also methods purposed for the (single-edit) multi-hop knowledge editing task, though not specifically evaluated

on MQuAKE in their original publications. We note that we are aware methods like GMeLLo [Anonymous, 2024], GLAME [Mengqi et al., 2024], RAE [Shi et al., 2024], StableKE [Wei et al., 2024], and Temple-MQA [Cheng et al., 2024] are also evaluated on MQuAKE, but they are purposely omitted from our re-benchmark coverage due to lack of open-sourced implementation, likely because most of these works are still in submission. Last, we note DeepEdit [Wang et al., 2024] is also an open-sourced MQuAKE-evaluated method, but we excluded it due to its lack of inference optimization (>200 A100 GPU hours needed for 1-edit on MQuAKE-Remastered-CF-3K).

**Covered Models**   We opt to use lmsys/vicuna-7b-v1.5 [Zheng et al., 2023b], mistralai/Mistral-7B-Instruct-v0.2 [Jiang et al., 2023], and meta-llama/Meta-Llama-3-8B-Instruct [AI@Meta, 2024] as the choice of question-answering models, both for alignment with existing works [Zhong et al., 2023, Shi et al., 2024, Gu et al., 2024] as well as providing coverage the most recent language models. For methods that require a text-embedding model as a retriever, we use facebook/contriever-msmarco [Izacard et al., 2022] for alignment with MeLLo [Zhong et al., 2023].

**Covered Datasets**   We will provide coverage on our post-fix dataset, namely MQuAKE-Remastered-CF, MQuAKE-Remastered-CF-3K, and MQuAKE-Remastered-T in the masking fashion illustrated in §4.2; as well as MQuAKE-Remastered-CF-6334 in its vanilla form. These datasets are respectively corresponding to the original MQuAKE-CF, MQuAKE-CF-3K, and MQuAKE-T from Zhong et al. [2023] (with 6334 as an extra for parameter-based methods), but with the types of error mentioned in §3 fixed in the via means illustrated in §4. We emphasize that such modification is legitimate, and our MQuAKE-Remastered is free for the scholarly community to adopt, as the original MQuAKE dataset was published under the MIT license. Where MQuAKE-Remastered will be released under CC BY 4.0. All experiments are conducted with an 80G NVIDIA A100 from a DGX A100 server.

## 5.2   Results and Discussion

Table 3: Performance Comparison of Original MQuAKE and our MQuAKE-Remastered datasets

| Method | MQuAKE-CF-3k | | MQuAKE-T | |
|---|---|---|---|---|
| | Original | Remastered | Original | Remastered |
| MeLLo [Zhong et al., 2023] | 6.7 | **6.77** | 30.84 | **44.37** |
| GWalk | 36.23 | **66.33** | 46.41 | **54.88** |

Table 4: Experiments on MQuAKE-Remastered-CF with numbers of edited cases and methods. Results inside ( ) are edited cases accuracy and unedited cases accuracy, respectively.

| Method | MQuAKE-Remastered-CF | | | | |
|---|---|---|---|---|---|
| | 1-edit | 1000-edit | 3000-edit | 6000-edit | 9171-edit |
| vicuna-7b-v1.5 [Zheng et al., 2023b] | | | | | |
| MeLLo [Zhong et al., 2023] | 22.55 (100, 22.54) | 21.54 (8, 23.2) | 17.79 (7.43, 22.83) | 12.62 (7.28, 22.58) | 6.95 (6.95, N/A) |
| ICE [Cohen et al., 2023] | <1 | OOM | OOM | OOM | OOM |
| IKE [Zheng et al., 2023a] | <1 | OOM | OOM | OOM | OOM |
| GWalk (Ours) | **61.89** (100, 61.89) | **56.98** (56.2, 57.07) | **56.37** (53.97, 57.54) | **54.93** (53.27, 58.06) | **54.15** (54.15, N/A) |
| Mistral-7B-Instruct-v0.2 [Jiang et al., 2023] | | | | | |
| MeLLo [Zhong et al., 2023] | 19.83 (<1, 19.84) | 19.08 (20.6, 18.9) | 18.9 (19.47, 18.62) | 18.27 (19.02, 16.87) | 18.09 (18.09, N/A) |
| ICE [Cohen et al., 2023] | <1 | OOM | OOM | OOM | OOM |
| IKE [Zheng et al., 2023a] | <1 | OOM | OOM | OOM | OOM |
| GWalk (Ours) | **61.42** (100, 61.42) | **57.79** (51.8, 58.52) | **56.35** (52.3, 58.32) | **53.73** (50.93, 59.04) | **51.53** (51.53, N/A) |
| Meta-Llama-3-8B-Instruct [AI@Meta, 2024] | | | | | |
| MeLLo [Zhong et al., 2023] | <1 | <1 | <1 | <1 | <1 |
| ICE [Cohen et al., 2023] | <1 | OOM | OOM | OOM | OOM |
| IKE [Zheng et al., 2023a] | <1 | OOM | OOM | OOM | OOM |
| GWalk (Ours) | **74.09** (100, 74.09) | **73.67** (71.1, 73.98) | **72.4** (70.9, 73.13) | **71.62** (70.33, 74.05) | **70.08** (70.08, N/A) |

Given our MQuAKE-Remastered are mostly provided as a fix to MQuAKE, we would like to first highlight the drastic results difference when the same method is evaluated on these two datasets. Table 3 shows our fixing can indeed result in drastically different experiment reports. Where such difference is especially significant for stronger methods, suggesting all previous reporting on MQuAKE has room for reliability improvements, which we filled here with MQuAKE-Remastered.

Due to page limitation, we only present the benchmark results on MQUAKE-REMASTERED-CF in the main text and refer our readers to Appendix D.2 for benchmarks of MQUAKE-REMASTERED-CF-3K, MQUAKE-REMASTERED-T, and MQUAKE-REMASTERED-CF-6334. Given the dominance of GWalk — a demo method we proposed as guidance to future scholars of this MHKE task — we leave more discussion on this method below.

### 5.3 Making Faithful Approach to MQUAKE and MQUAKE-REMASTERED

Additionally, it is also our observation that many multi-hop knowledge editing methods with decent accuracy reports on MQUAKE or MQUAKE-REMASTERED are utilizing designs that leverage specific data properties unique to MQUAKE. For example, methods like GLAME [Mengqi et al., 2024] utilize Wikidata [Vrandečić and Krötzsch, 2014] as the external knowledge graph to better detect the edit-induced conflicts, which happen to be the source of MQUAKE as discussed in §2.1. While these methods might have decent performance on MQUAKE, the cost of maintaining a positive knowledge graph on the correct — but not just edited — knowledge facts is undoubtedly a non-trivial operation cost. Yet, whether sourcing the same Wikidata knowledge graph as MQUAKE might bring them data-specific advantages remains unanswered. Similarly, PokeMQA [Gu et al., 2024] utilizes the 6,218 cases included in MQUAKE-CF but not in MQUAKE-CF-3K as the train set to train its auxiliary components. Given MQUAKE is a dataset with relatively low diversity (e.g., it only includes 37 types of relations), whether having a heavily overlapped train and test set will result in data-specific advantages unique MQUAKE and its variants, again remains unanswered.

**A Minimally Invasive but Performant Approach: GWalk**   Here, we provide a brief walkthrough of a simple method we designed, namely GraphWalk. It does not leverage any data-specific property unique to MQUAKE or MQUAKE-REMASTERED, yet still presents pleasant performance surpassing many established baselines. We illustrate this method as a simple guidance and potential inspiration to our future multi-hop knowledge editing scholars. **Due to page limitation, we introduce the technical details and design intuition of GWalks in Appendix D.1**, and only present the performance of GWalks in the main text.

We hope the performant nature of GWalk — in its most vanilla form, without employing any data-specific property unique to MQUAKE or MQUAKE-REMASTERED — can inspire more multi-hop knowledge editing methods that leverage the graph topology of edited facts, without converting such facts to natural language descriptions (at least for retrieval).

## 6 Related Works

Our work mainly conducts an audit and provides fixes to the MQUAKE dataset. To the best of our knowledge, only two prior arts have touched on the errors existing in MQUAKE: GMeLLo [Anonymous, 2024] (an anonymous submission to ACL ARR 2024 February) and DeepEdit [Wang et al., 2024]. As an overview, GMeLLo briefly discussed the same type of error we discussed in §3.4 without providing any quantitative error analysis or any fix. DeepEdit discovered the same inner contamination error as we discussed in §3.2, but specific to *3000-edit* setup. DeepEdit's proposed fix is simply removing the 998 inner contaminated cases from the MQUAKE-CF-3K dataset, so this fix is custom 3000-edit and done so by sacrificing 1/3 of the dataset capacity. We leave more details in Appendix E due to page limitation.

Additionally, our work provides a re-benchmark of most, if not all, open-sourced knowledge editing methods evaluated on MQUAKE, and sets guidance on how to faithfully approach such datasets. To the best of our knowledge, no other work provides the same benchmark nor touches on the same issue.

## 7 Conclusion

Our work provides a comprehensive audit and fix of the MQUAKE dataset. We further re-benchmarked all open-sourced knowledge editing methods evaluated on MQUAKE with our MQUAKE-REMASTERED datasets and provided guidance and examples on how to faithfully approach these datasets with our GWalk.

## Limitations and Impact Statement

While our work comprehensively addressed many errors in MQUAKE, we caution our reader to perform further analysis and evaluation on our MQUAKE-REMASTERED to ensure our fixes are indeed exhaustive. We also note that multi-hop knowledge editing only represents one aspect of a language model's ability, so any actual deployment of a language model should undergo more, and if possible, deployment-specific evaluations.

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

# A  Extended Preliminary

## A.1  Demo Report of MQUAKE

Table 5: Standard reporting format of MQUAKE-CF-3K, and MQUAKE-T demoed with MeLLo on Vicuna-7B [Zheng et al., 2023b]; $k$-edited means $k$ cases out of the total cases are edited. Abbreviated table courtesy of Zhong et al. [2023] (Table 3).

| Model | Method | MQUAKE-CF-3K | | | | MQUAKE-T | | | |
|---|---|---|---|---|---|---|---|---|---|
| | | 1-edit | 100-edit | 1000-edit | 3000-edit | 1-edit | 100-edit | 500-edit | 1868-edit |
| Vicuna-7B | MeLLo [Zhong et al., 2023] | 20.3 | 11.9 | 11.0 | 10.2 | 84.4 | 56.3 | 52.6 | 51.3 |

## A.2  Dataset Statistics

Table 6: Dataset Statistics of MQUAKE. Numbers are in terms of cases (a case in MQUAKE is a chain consisting of multiple subquestions).

| Dataset | # of Edits | 2-hop | 3-hop | 4-hop | Total |
|---|---|---|---|---|---|
| MQUAKE-CF-3K | 1 | 513 | 356 | 224 | 1,093 |
| | 2 | 487 | 334 | 246 | 1,067 |
| | 3 | - | 310 | 262 | 572 |
| | 4 | - | - | 268 | 268 |
| | All | 1,000 | 1,000 | 1,000 | 3,000 |
| MQUAKE-CF | 1 | 2,454 | 855 | 446 | 3,755 |
| | 2 | 2,425 | 853 | 467 | 3,745 |
| | 3 | - | 827 | 455 | 1,282 |
| | 4 | - | - | 436 | 436 |
| | All | 4,879 | 2,535 | 1,804 | 9,218 |
| MQUAKE-T | 1 (All) | 1,421 | 445 | 2 | 1,868 |

Table 7: Dataset Statistics of MQUAKE-REMASTERED. Numbers are in terms of cases (a case in MQUAKE is a chain consisting of multiple subquestions).

| Dataset | # of Edits | 2-hop | 3-hop | 4-hop | Total |
|---|---|---|---|---|---|
| MQUAKE-REMASTERED-CF-3K | 1 | 513 | 356 | 224 | 1,093 |
| | 2 | 487 | 334 | 246 | 1,067 |
| | 3 | - | 310 | 262 | 572 |
| | 4 | - | - | 268 | 268 |
| | All | 1,000 | 1,000 | 1,000 | 3,000 |
| MQUAKE-REMASTERED-CF | 1 | 2,446 | 850 | 441 | 3,737 |
| | 2 | 2,415 | 852 | 463 | 3,730 |
| | 3 | - | 823 | 451 | 1,274 |
| | 4 | - | - | 430 | 430 |
| | All | 4,861 | 2,525 | 1,785 | 9,171 |
| MQUAKE-REMASTERED-T | 1 (All) | 1,421 | 441 | 2 | 1,868 |
| MQUAKE-REMASTERED-CF-6334 | 1 | 1,971 | 77 | 0 | 2,048 |
| | 2 | 2,415 | 476 | 14 | 2,905 |
| | 3 | - | 823 | 128 | 951 |
| | 4 | - | - | 430 | 430 |
| | All | 4,386 | 1,376 | 572 | 6,334 |

## B  Extended Auditing

### B.1  Example of Inner Contamination between Different Edited Cases (§3.2)

Again, we walk through two cases from MQUAKE-CF-3K as a concrete example. First, we show them in their unedited format (again, subquestion breakdowns and intermediate answers are here for demonstration purposes and are not visible to the question-answering LLM during evaluation):

- `case_id:1570` (unedited): *Who was the creator of the official language used in the work location of Matti Vanhanen?*
  - ◇ Which city did Matti Vanhanen work in? Helsinki.
  - ◇ What is the official language of Helsinki? Finnish.
  - ◇ Who was Finnish created by? Mikael Agricola.

- `case_id:1968` (unedited): *Who created the official language of Housemarque's headquarters location?*
  - ◇ Which city is the headquarter of Housemarque located in? Helsinki.
  - ◇ What is the official language of Helsinki? Finnish.
  - ◇ Who was Finnish created by? Mikael Agricola.

Suppose `case_id:1570` and `case_id:1968` are both selected as editing cases, two triples containing the following knowledge will be available: *"The official language of Helsinki is Black Speech"* (intended for `case_id:1570`), and *"Finnish was created by William Shakespeare"* (intended for `case_id:case_id:1968`), leading to the following edited breakdown.

- `case_id:1570` (edited): *Who was the creator of the official language used in the work location of Matti Vanhanen?*
  - ◇ Which city did Matti Vanhanen work in? Helsinki.
  - ◇ What is the official language of Helsinki? ~~Finnish~~ Black Speech.
  - ◇ Who was ~~Finnish~~ Black Speech created by? J. R. R. Tolkien.
- `case_id:1968` (edited): *Who created the official language of Housemarque's headquarters location?*
  - ◇ Which city is the headquarter of Housemarque located in? Helsinki.
  - ◇ What is the official language of Helsinki? Finnish.
  - ◇ Who was Finnish created by? ~~Mikael Agricola~~ William Shakespeare.

Much like the previous conflict between unedited and edited cases, these two edited cases share a common subquestion: *"What is the official language of Helsinki?"* However, such subquestion is edited in `case_id:1570` while unedited in `case_id:1968`, causing unintended contamination.

## B.2 Error Analysis of MQUAKE-CF

Table 8: Error statistics of MQUAKE-CF [Zhong et al., 2023] in terms of edited cases contaminating unedited cases §3.1. $k$-edited means $k$ cases are edited out of the total 9218 cases.

| # of Contaminated | MQUAKE-CF-3K | | | | | | |
|---|---|---|---|---|---|---|---|
| | 1-edit | 100-edit | 1000-edit | 2000-edit | 3000-edit | 5000-edit | 9218-edit |
| Cases | 62 | 3307 | 5275 | 5110 | 4578 | 3346 | 0 |
| Subquestions | 62 | 4525 | 8751 | 8989 | 8326 | 6364 | 0 |

Table 9: Error statistics of MQUAKE-CF [Zhong et al., 2023] in terms edited cases contaminating each others §3.2. $k$-edited means $k$ cases are edited out of the total 9218 cases.

| # of Contaminated | 1-edit | 100-edit | 1000-edit | 2000-edit | 3000-edit | 5000-edit | 9218-edit |
|---|---|---|---|---|---|---|---|
| Cases | 0 | 8 | 192 | 441 | 732 | 1397 | 2873 |
| Subquestions | 0 | 12 | 270 | 606 | 1027 | 1986 | 4250 |

## C   Extended Remastering

### C.1   Contamination Free Subset: MQUAKE-REMASTERED-CF-6334

While MQUAKE-REMASTERED-MASKED with masking operation can well support memory-based knowledge editing methods, it will not be compatible with parameter-based methods. This is because, for parameter-based methods, the set of edited facts used for training and evaluation needs to be constant yet consistent with each other at all times; whereas dynamic masking cannot suffice as it is essentially adjusting the dataset on the fly during inference time.

To effectively evaluate parameter-based knowledge editing methods, we present MQUAKE-REMASTERED-CF-6334. MQUAKE-REMASTERED-CF-6334 is a dataset extracted from MQUAKE-CF, where all 6,334 cases are edited cases; and they are completely contamination-free from each other. This dataset is suitable for LLM editing with parameter-based approaches, as one can make careful splits among the 6,334 cases of MQUAKE-REMASTERED-CF-6334 to serve as train, validation, and evaluation sets.

# D Extended Benchmark and Discussion

## D.1 GWalks

The design of GWalk hinges on the fundamental pipeline of memory-based knowledge editing methods: where the pool of source only contains *edited facts*. This school of editing methods has proven to be successful, mainly because it can leverage the power of retrieval-argument generation (RAG) combined with the in-context learning (ICL) capability of LLMs. Further, it is common sense that edited knowledge facts will be much less than unedited knowledge facts, making maintaining a knowledge pool exclusively containing edited facts a viable option — like done so in MeLLo [Zhong et al., 2023].

Different from MeLLo, where all edited facts are converted from triples to natural language (NL) descriptions in its edited bank, GWalk preserves the edited facts in their original triples fashion and leverages the graph topology they come with. This makes maintaining this edited bank much easier — as one can easily adjust the entity or relation on a knowledge graph without rewriting every natural language description of every related edited fact. It also brings more precise retrieval mapping when a question pertaining to a certain edited fact is asked. This is because methods like MeLLo would need to RAG from a pool of edited facts in NL format, and there might always be something — though not actually related to the question asked — having a close enough embedding distance to the query question (i.e., unintended retrieval), and thus result in hallucination. However, if we simply query the entity and relations implied in a question against a knowledge graph, there is less chance of retrieving unintended materials. Specifically, GWalk works like the following Algorithm 1.

---

**Algorithm 1:** General Procedure GWalk on a Multi-hop Question

---

**Input:**
    $M$, the Question Answering Language Model;
    $T$, a Text-embedding model;
    $Q$, a Multi-hop Question;
    $E$, a bank of edited facts as a knowledge graph.

**Output:**
    $o_p$, the answer to $Q$.

**Initialize:**
    $i = 1$, the subquestion counter;
    $o_p =$ None, the answer from the previous subquestion.

1   $s \leftarrow$ Extracted subject from $Q$;
2   $rels \leftarrow$ Prompt $M$ to breakdown $Q$ into a sequence of relations.

```
/* If Q is 'What is the official language of the country where Karl
   Alvarez holds citizenship?', then s would be 'Karl Alvarez' and a
   possible rels is ['citizenship', 'official language']          */
```

3   **for** $r \in rels$ **do**
4      Query $< s, r, ? >$ against $E$ using $T$, namely we do $T(s)$ first to determine if there is a retrievable $s \in E$, then inspect if the $s \in E$ has an relation edge retrievable by $T(r)$.

```
   /* We set a threshold on embedding similarity for T to determine
      whether an item is retrievable or not.                      */
```

5      Prompt $M$ to generate subquestion $q_i$ with $s$ and $r$.
6      $o_p \leftarrow$ the $M$-generated answer to $q_i$.
7      **if** $T(s, r)$ has a valid retrieval $< s, r, o^* >$ **then**
8         $o_p \leftarrow o^*$;

```
   /* The answer to this subquestion will be the start subject of the
      next subquestion.                                            */
```

9      $s \leftarrow o_p$ ;
10      $i \leftarrow i + 1$;

11   **Return** $o_p$;

---

 **D.2 Additional Experiment Results**

Table 10: MQUAKE-REMASTERED-CF-3K

| Method | MQUAKE-REMASTERED-CF-3K | | | |
| --- | --- | --- | --- | --- |
| | **1-edit** | **100-edit** | **1000-edit** | **3000-edit** |
| vicuna-7b-v1.5 [Zheng et al., 2023b] | | | | |
| MeLLo [Zhong et al., 2023] | 16.54 (100, 16.51) | 18 (9.0, 18.31) | 14.63 (8.0, 17.95) | 6.77 (6.77, N/A) |
| ICE [Cohen et al., 2023] | <1 | <1 | OOM | OOM |
| OOM | | | | |
| IKE [Zheng et al., 2023a] | <1 | OOM | OOM | OOM |
| OOM | | | | |
| GWalk (Ours) | **54.89** (100, 54.87) | **60.9** (54, 61.14) | **57.37** (54.4, 58.85) | **66.33** (66.33, N/A) |
| Mistral-7B-Instruct-v0.2 [Jiang et al., 2023] | | | | |
| MeLLo [Zhong et al., 2023] | 19.73 (100, 19.71) | 18.6 (21, 18.52) | 16.33 (17.8, 15.6) | 15.93 (15.93, N/A) |
| ICE [Cohen et al., 2023] | <1 | <1 | OOM | OOM |
| OOM | | | | |
| IKE [Zheng et al., 2023a] | <1 | 4.43 (4,4.49) | OOM | OOM |
| OOM | | | | |
| GWalk (Ours) | **56.57** (100, 56.55) | **61.93** (47, 62.45) | **57.17** (51.5, 60.0) | **51.0** (51.0, N/A) |
| Meta-Llama-3-8B-Instruct [AI@Meta, 2024] | | | | |
| MeLLo [Zhong et al., 2023] | <1 | <1 (2.0, <1) | 1.03 (3.0, <1) | 2.3 (2.3, N/A) |
| ICE [Cohen et al., 2023] | <1 | <1 | OOM | OOM |
| OOM | | | | |
| IKE [Zheng et al., 2023a] | <1 | <1 | OOM | OOM |
| OOM | | | | |
| GWalk(Ours) | **69.0** (100, 68.99) | **76.73** (67, 77.07) | **75.47** (74.2, 76.1) | **70.6** (70.6, N/A) |

*Results inside the parenthesis are edited cases accuracy and unedited cases accuracy, respectively.

Table 11: MQUAKE-REMASTERED-T

| Method | MQUAKE-REMASTERED-T | | | |
| --- | --- | --- | --- | --- |
| | **1-edit** | **100-edit** | **500-edit** | **1864-edit** |
| vicuna-7b-v1.5 [Zheng et al., 2023b] | | | | |
| MeLLo [Zhong et al., 2023] | 19.31 (100, 19.27) | 18.88 (45.0, 17.4) | 22.16 (40.4, 15.47) | 44.37 (44.37, N/A) |
| ICE [Cohen et al., 2023] | <1 | <1 | <1 | OOM |
| IKE [Zheng et al., 2023a] | <1 | <1 | <1 | OOM |
| GWalk (Ours) | **35.52** (100, 35.48) | **46.51** (49.0, 46.37) | **48.93** (56.0, 46.33) | **54.88** (54.88, N/A) |
| Mistral-7B-Instruct-v0.2 [Jiang et al., 2023] | | | | |
| MeLLo [Zhong et al., 2023] | 10.3 (0, 10.31) | 10.25 (59.0, 7.48) | 18.78 (48.4, 7.92) | 47.75 (47.75, N/A) |
| ICE [Cohen et al., 2023] | <1 | <1 | <1 | OOM |
| IKE [Zheng et al., 2023a] | <1 | <1 | <1 | OOM |
| GWalk (Ours) | **34.07** (0, 34.08) | **45.76** (47, 45.69) | **46.78** (51.2, 45.16) | **50.7** (50.7, N/A) |
| Meta-Llama-3-8B-Instruct [AI@Meta, 2024] | | | | |
| MeLLo [Zhong et al., 2023] | <1 | 1.13 (17, 0.23) | 4.72 (17.4, <1) | 16.58 (16.58, N/A) |
| ICE [Cohen et al., 2023] | <1 | <1 | <1 | OOM |
| IKE [Zheng et al., 2023a] | <1 | <1 | <1 | OOM |
| GWalk (Ours) | **70.12** (100, 70.1) | **73.28** (84.0, 72.68) | **76.61** (87, 72.8) | **84.01** (84.01, N/A) |

*Results inside the parenthesis are edited cases accuracy and unedited cases accuracy, respectively.

Table 12: MQ​uAKE-Remastered-cf-6334

| Method | MQuAKE-Remastered-cf-6334 | | | |
|---|---|---|---|---|
| | **100-edit** | **1000-edit** | **3000-edit** | **6344-edit** |
| vicuna-7b-v1.5 [Zheng et al., 2023b] | | | | |
| MeLLo [Zhong et al., 2023] | 19.16
(0, 10.99, 19.37) | 19.27
(5.1, 9.58, 24.53) | 11.17
(4.31, 8.55, 23.3) | 6.83
(4.58, 7.72, 19.05) |
| ICE [Cohen et al., 2023] | OOM | OOM | OOM | OOM |
| IKE [Zheng et al., 2023a] | OOM | OOM | OOM | OOM |
| PokeMQA [Gu et al., 2024] | - | - | - | 21.77
(3.25, 30.82, 1.59) |
| GWalk (Ours) KGWalk | **57.55**
(22.22, 64.84, 57.48) | **61.79**
(29.08, 66.17, 63.23) | **59.1**
(39.3, 63.74, 64.33) | **56.62**
(44.64, 62.11, 68.25) |
| Mistral-7B-Instruct-v0.2 [Jiang et al., 2023] | | | | |
| MeLLo [Zhong et al., 2023] | 27.5
(<1, 23.08, 27.65) | 27.54
(12.76, 24, 30.4) | 24.37
(11.88, 25.51, 32.06) | 21.26
(13.29, 24.9, 30.16) |
| ICE [Cohen et al., 2023] | OOM | OOM | OOM | OOM |
| IKE [Zheng et al., 2023a] | 8.82
(11.11,6.59,8.86) | OOM | OOM | OOM |
| PokeMQA [Gu et al., 2024] | - | - | - | 20.38
(3.99, 27.41, 69.84) |
| GWalk (Ours) | **56.25**
(33.33, 57.14, 56.28) | **58.9**
(34.69, 60.57, 60.6) | **56.03**
(42.69, 59.04, 59.85) | **54.43**
(47.49, 57.74, 52.38) |
| Meta-Llama-3-8B-Instruct [AI@Meta, 2024] | | | | |
| MeLLo [Zhong et al., 2023] | <1 | <1 | 1.12
(1.17, 1.48, 0.22) | 1.27
(<1, 1.4, 1.59) |
| ICE [Cohen et al., 2023] | OOM | OOM | OOM | OOM |
| IKE [Zheng et al., 2023a] | <1 | OOM | OOM | OOM |
| PokeMQA [Gu et al., 2024] | - | - | - | 20.38
(1.08, 28.41, 76.19) |
| GWalk (Ours) | **67.01**
(33.33, 74.73, 66.92) | **71.89**
(47.45, 80.94, 70.65) | **73.76**
(54.05, 81.6, 71.12) | **74.22**
(61.02, 80.47, 73.02) |

*Results inside the parenthesis are edited cases (unique in the test set) accuracy, edited cases (overlap of the test and train set) accuracy, and unedited cases accuracy, respectively.

# E  Extended Related Works

Specifically, GMeLLo [Anonymous, 2024] briefly discusses the inconsistency between the triple chain and the generated multi-hop questions in its §4.5.1, which is the same type of error we discussed in §3.4. We note that GMeLLo merely highlights such errors but does not provide a quantified measurement of its scale nor any fix. We did both in §3.4 and §4.1.

DeepEdit [Wang et al., 2024] discovered the same inner contamination error as we discussed in §3.2. DeepEdit does provide a quantified measurement of the scale of such error but only pertains to the MQUAKE-CF-3K dataset, and such quantifiable results are only valid when all 3,000 cases of MQUAKE-CF-3K are considered edited; which, as shown in Table 5, only constitute one column of MQUAKE-CF-3K's reporting. Further, DeepEdit provides a rather hardcore fix to this problem by removing the 998 inner contaminated cases from the MQUAKE-CF-3K dataset — which is (supposedly) the same 998 cases we detect in Table 2 under the 3000-edit column — with the post-fix dataset denoted as MQUAKE-2002 for having 2,002 out of 3,000 cases left. While this fix is, of course, helpful, we argue our post-fix MQUAKE-REMASTERED-CF-3K, MQUAKE-REMASTERED-CF, and MQUAKE-REMASTERED-T are much more comprehensive and effective since they patched many more errors revealed in §3 (which still exists in MQUAKE-2002), works outside the MQUAKE-CF-3K dataset, do not require the number of edits to be 2,002 cases, and most importantly, done so without scarifying almost 1/3 of the capacity of the original dataset.

