# OpenReview forum: "MQuAKE-Remastered: Multi-Hop Knowledge Editing Can Only Be Advanced With Reliable Evaluations"
_NeurIPS.cc/2024/Datasets_and_Benchmarks_Track — Submitted to NeurIPS 2024 Track Datasets and Benchmarks_

### Official Review · Reviewer_ujS8 · 2024-07-25

**Rating:** 6
**Confidence:** 4
**Correctness:** Yes
**Clarity:** No. Refer to the cons (2) above for d…

**Review:**

Pros: This paper makes a very solid contribution on revisiting the pitfalls hidden in an established benchmark and its evaluation practices. I have no doubt that this paper should be followed and used in future works on knowledge editing.

Cons:

1. Considering that this paper only revisits MQuAKE dataset, which is not the only dataset on its topic (RippleEval of Cohen et al., 2023 can do essentially the same thing), its audiences could be somehow limited.

2. The presentation of this paper has significant room for improvement. (1) The writing style overusing bold texts and section titles makes the paper somehow messy and a little bit hard to follow. (2) The presentation of the experimental results (Table 3 and 4) makes them hard to interpret.

**Strengths:**

Please refer to the pros above.

**Additional Feedback:**

Regarding the dynamic masking method described in section 4.2, is it possible to provide a fixed split of MQuAKE avoiding the knowledge overlap of edited and unedited knowledge? In this way, the dynamic masking technique is unnecessary.

**Documentation:**

The URL provided to access the dataset is unavailable.

**Limitations:**

Yes

**Opportunities For Improvement:**

Please refer to the cons above.

**Relation To Prior Work:**

Yes

**Summary And Contributions:**

This paper presents a comprehensive audit and correction of the MQUAKE dataset, which is widely used for evaluating multi-hop knowledge editing methods in large language models. The authors identify and fix significant errors in MQUAKE, including contamination between edited and unedited cases, conflicting edits, and missing information in question instructions, etc. The paper also provides new benchmarking results.

---

> ### Author Rebuttal · Authors · 2024-08-15
>
> ## **[W1 - RippleEdit does the same thing as MQuAKE]: Not really. RippleEdit is a single-edit dataset, whereas MQuAKE is multi-edit. To the best of our knowledge, MQuAKE is the only available multi-edit multi-hop knowledge editing dataset by design as of today.**
>
> RippleEdit by [1] is indeed a multi-hop editing dataset like MQuAKE, but it is single-edit in nature. This means that at any given moment, the editor's edited knowledge bank has only one piece of edited fact, evidenced by:
>
> > `RippleEdit Section 6`: *"In addition, while we focus on ripple effects of single edits, future work can consider the effect of editing multiple facts in a single batch."*
>
> So RippleEdit, by its own design, essentially only cares about whether a method can a) retrieve the edited fact at the right moment and b) interpret the edited fact correctly to form a correct, editing-following answer. **This is different from the MQuAKE's setup, where multiple edited facts can co-exist in the same edited knowledge bank,  and one of the most significant challenges is to retrieve the relevant edited fact upon a certain editing-influenced subquestion.**
>
> We consider MQuAKE's task design and setup to be more reflective of real-world editing tasks, as naturally, there will always be more than one edited fact stored for any system with reasonable complexity. That being said, we are happy to report our proposed pilot method, GWalk, performs decently on RippleEdit. Here are some snapshot results on Llama-2-7b-chat:
>
> > Single-edit result of RippleEdit-Popular/Recent/Random. C1/2 means the edit is happening at the 1st or the 2nd hop (RippleEdit cases only have 2 hops).
>
>
> | Method | Popular C1 Acc. | Popular C2 Acc. |  Recent C1 Acc. | Recent C2 Acc. |  Random C1 Acc. | Random C2 Acc. |
> |-|-|-|-|-|-|-|
> | ROME | 37.4 | 16.2 |  47.8 | 50.0 | 35.5 | 49.5 |
> | ICE | 85.1 | 67.6 | 74.8 | 85.0 |  73.8 | 80.3 |
> | MeLLo | 45.1 | 77.1 | 50.2 | 80.0 |  40.2 | 68.3 |
> | GWalk (ours) | **85.7** | **81.8** | **80.9** | **87.6** | **76.1** | **82.9** |
>
>
> ---
> We additionally convert RippleEdit to a multi-edit setup — i.e., there are multiple edited facts within the editing knowledge bank at the same time — to a) make it more challenging and, b) show that our audit can also *"fix"* issues within a different dataset. Note we put the "fix" in quotes as RippleEdit is not designed with multi-edit in mind, so the things we fixed are not necessarily errors but just some adjustments required for making a proper multi-edit dataset. In any case, here are the snapshot results on Llama-3-8b-Instruct:
>
> > Multi-edit result of RippleEdit-Popular/Recent/Random. In this case, we fixed 21/2/0 conflict edits and 3/120/1 case-to-case contamination within RippleEdit-Ropular/Recent/Random datasets, respectively.
>
> | Method | Popular C1 Acc. | Popular C2 Acc. |  Recent C1 Acc. | Recent C2 Acc. |  Random C1 Acc. | Random C2 Acc. |
> |-|-|-|-|-|-|-|
> | MeLLo | 35.1 | 40.3 | 41.1 | 42.4 | 49.5 | 50.0 |
> | GWalk (ours) | **79.0** | **66.9** | **79.2** | **63.9** | **72.9** | **60.0** |
>
>
>
> ---
>
> ## **[W2.1 - Overusing bold texts and section titles: Sure, we will tone down a few notches.**
>
> ---
> ## **[W2.2 - Experiment results are hard to interpret]: Sure, we will highlight/format them better.**
>
> We believe the reviewer's concerns are mostly related to Table 4-like results, where we present "Total Acc (Edited Cases Acc, Unedited Cases Acc)" in the same cell. We will break them off into different columns in the future and highlight them in the table header.
>
> We are not too sure how Table 3 is hard to interpret, as this is just a small before-and-after fix comparison table. We will highlight what we are trying to deliver in this table in its caption (basically showing that fixing the error will yield a very different performance report on the same method, thus all exisiting original MQuAKE-based results are unreliable), which should help for digestion. Please do let us know if there is something else we need to address — we are all ears.
>
>
> ---
>
> ## **[Documentation - URL link for the dataset is unavailable]: The GitHub repo is currently private pending Meta's legal review; the exact copy of the repo is submitted as a zip file.**
>
> We thank the reviewer for being meticulous. As specified in our supplied `datasheet.pdf` under supplementary material, currently, the dataset is under Meta's legal review. But the same exact copy of dataset, metadata, code, documentation, etc are submitted as a zip file as supplementary material. D&B calling specified that this is allowed as long as everything is released by camera-ready, which we certainly plan to do so.
>
>
> ---
>
> [1] Cohen et al., Evaluating the Ripple Effects of Knowledge Editing in Language Models. TACL 2024

---

> ### Author Response · Authors · 2024-08-21
> **Additional results on DeepEdit and RAE.**
>
> As highlighted around `L289 - Compared Methods` paragraph, our work featured almost all (-1) reproducible multi-hop knowledge editing methods evaluated on MQuAKE. Unfortunately, there are many more methods we cannot feature due to a lack of open-source implementations, potentially due to most of these methods are still under submission.
>
> We are glad to work with RAE — a recently accepted multi-hop editing method to CIKM, which was also evaluated on MQuAKE — core authors and provided the following pilot results, in addition to our recently obtained DeepEdit reports. **The reviewer didn't ask for any of it, but we thought it might be a nice thing to add — as not only the following results further strengthen the significant performance advantage of our GWalk method, but it also makes our work the [most comprehensive evaluation](https://openreview.net/forum?id=iTUlYblV0K&noteId=TvSHM0uGhA) of multi-hop knowledge editing with a healthy margin over the runner-up.**
>
>
> > DeepEdit, RAE, and our GWalk on MQuAKE-Remastered-CF-6334 with Llama-3-8B-Instruct, for four different #-edit settings. The following results are obtained under the same setup as Table 12, so they are directly comparable. Results on more models/datasets will be added in our updated manuscript.
> >
> > Header Definition:
> > **#-Edit**: # of cases are considered edited;
> > **Test Edited**: Acc. of edited cases uniquely exist in the test set;
> > **Overlapped Edited**: Acc. of edited cases that exist both in the train and test sets (this metric is unique to the CF-6334 dataset, as this dataset is made with parameter-based methods in mind, where a consistant and constant knowledge base between train and test set is required. More on this in Appendix C.1);
> > **Test Unedited**: Acc. of unedited cases exist in the test set.
> > We additionally note the originally RAE design extracts few-shot examples from {MQuAKE-CF(-9K) - MQuAKE-CF-3K}, we specifically turned off this step due to the low diversity concern between two datasets, as discussed around `L338`, as well as to be aligned with other Table 12-featured methods, like PokeMQA.
>
>
> |Method|#-Edit|Test Edited|Overlapped Edited|Test Unedited|Overall Acc.|
> |-|-|-|-|-|-|
> |DeepEdit|100|11.1|19.78|24.29|24.13|
> |RAE|100|22.2|12.09|29.74|29.33|
> |GWalk (ours)|100|**33.3**|**74.73**|**66.92**|**67.01**|
>
>
>
>
>
>
> |Method|#-Edit|Test Edited|Overlapped Edited|Test Unedited|Overall Acc.|
> |-|-|-|-|-|-|
> |DeepEdit|1000|8.16|20.52|26.27|24.35|
> |RAE|1000|33.67|11.67|32.49|25.65|
> |GWalk (ours)|1000|**47.45**|**80.94**|**70.65**|**71.89**|
>
>
>
>
>
> |Method|#-Edit|Test Edited|Overlapped Edited|Test Unedited|Overall Acc.|
> |-|-|-|-|-|-|
> |DeepEdit|3000|7.57|19.65|25.38|21.01|
> |RAE|3000|23.11|10.12|33.48|15.59|
> |GWalk (ours)|3000|**54.05**|**81.6**|**71.12**|**73.76**|
>
>
> |Method|#-Edit|Test Edited|Overlapped Edited|Test Unedited|Overall Acc.|
> |-|-|-|-|-|-|
> |DeepEdit|6334|7.48|18.81|28.57|18.90|
> |RAE|6334|18.75|11.39|28.57|11.58|
> |GWalk (ours)|6334|**61.02**|**80.47**|**73.02**|**74.22**|
>
> Cheers,
> *Paper840 Authors*
>
> ---
> DeepEdit: DeepEdit: Knowledge Editing as Decoding with Constraints. arXiv 2024
> RAE: Shi et al., Retrieval-Enhanced Knowledge Editing for Multi-Hop Question Answering in Language Models. CIKM 2024

---

> ### Author Response · Authors · 2024-08-23
> **An invitation to discuss, as we believe we have addressed the RippleEdits concerns in all ways possible.**
>
> Dear Reviewer `ujS8`,
>
> Outside some presentation adjustments, your only other initial concern is our work's lack of coverage on other multi-hop knowledge editing datasets, in which you correctly pointed out RippleEdits by Cohen et al., as a potential alternative.
>
> In our [previous rebuttal](https://openreview.net/forum?id=iTUlYblV0K&noteId=qSbeth2p55), we have analyzed that **RippleEdits is, in fact, not like MQuAKE**. RippleEdits is a single-edit dataset where at any given moment, the editor's edited knowledge bank has only one piece of edited fact (in contrast to MQuAKE's multi-edit setting,
> where multiple edited facts can co-exist in the same edited knowledge bank). **Given there will be no contamination with just one edited fact, much of our fixing are not applicable to RippleEdits in its original single-edit setting**. We believe it is clear that MQuAKE's setting is more challenging and reflective of real-world editing scenarios, as obviously, any system would have more than one mistake to edit. And as far as we know, MQuAKE is the *ONLY* publically available multi-hop editing dataset designed under this reflective setting.
>
> Further, we have conducted **additional evaluations on RippleEdits.** Both under its original single-edit setting, as well as converting it to a multi-edit setup (and conducting additional "fixing" to ensure there are no contaminations — we put it in quotes as this is different from RippleEdits' original design, so what we "fixed" are not actually "mistakes" in the first place, but more of some adjustment to make RippleEdits multi-edit compatible). From our perspective, we believe we have done everything possible to address these RippleEdits-centroid concerns, and we hope our reviewer will let us know if the message is well-received at your end. Thanks in advance.

---

### Official Review · Reviewer_wp4u · 2024-07-26
**a comprehensive audit and fix of the MQUAKE dataset, which is commonly used to evaluate multi-hop knowledge editing methods for large language models**

**Rating:** 3
**Confidence:** 4
**Correctness:** yes
**Clarity:** yes

**Review:**

# Strengths

1.  The authors provide a detailed analysis of various errors in the original dataset, quantifying their impact. For example, they show that up to 76% of cases in MQUAKE-T can be contaminated with just 100 edited cases (Table 1).

2.  MQUAKE-REMASTERED addresses multiple types of errors, including hard corrections for conflicting edits, missing information, and duplicated cases. The authors also introduce dynamic masking to handle contamination issues.

3.   The paper evaluates several open-source knowledge editing methods on the new dataset, providing more reliable performance metrics. This is valuable for the research community to better understand the true capabilities of these methods.

4.  The authors propose a simple yet effective method that outperforms existing baselines on MQUAKE-REMASTERED. This demonstrates that complex, data-specific approaches may not be necessary for good performance.

5.  The authors provide detailed information about their experimental setup, including model versions and computing resources used, which aids in reproducibility.


# Post-Rebuttal Update on Opportunities For Improvement

I appreciate the rebuttal from the authors. The following concerns remain. I strongly suggest another round of revision to reach the bar of publication at one top-tier conference.

1. First of all, multi-hop editing is a well-studied field [1-5]. Most of the highly related multi-hop editing benchmarks are not fully discussed in the paper. Since Multi-hop editing is extensively benchmarked in literature, it is strongly suggested that the authors conduct another round of revision to thoroughly discussed the differences compared to all the highly related references [1-5] rather than only [1]. The current version only focuses on discussing the differences with [1], which is not enough for a well-studied field.
2. The motivation is flawed if the authors did not discuss the differences with all the the highly related benchmarks. The authors emphasized too much about one work [1] (e.g., “MQUAKE is flawed due to unintentional clerical and procedural errors “) and ignore the differences with other highly related multi-hop editing benchmarks.
3. The paper structure need to be improved dramatically. The authors use the whole section 2 to explain the MQUAKE benchmark but ignore the other recent multi-hop editing benchmarks. A comprehensive review of the existing benchmarks on multi-hop editing is desired.
4. Related works are highly incomplete and insufficient. Highly related works such as [3,6] are not referenced in this paper. Section 6 should be rewritten to thoroughly discuss the differences with all the highly related references [1-5]. The current version of related work section is no enough.
5. The contribution is limited. The authors only fix the errors in one dataset. No particularly significant insights are concluded.
6. Not much guidance is provided for future multi-hop editing model design. The authors briefly discussed on approach GWalk. No details are provided. It is strongly suggested the authors move the technical content from appendix to the main paper. Page limit is no excuse to put important technical details in the appendix
7. Not much insight can be found. It is strongly suggested that the authors summarize and highlight their insights better. It is hard to find valuable insights from the the current version.
8. The benchmarked methods are highly limited. Based on Table 4, ONLY three existing methods are benchmarked, and most results are OOM. Table 4 actually contains highly limited information.
9. The experiments are highly limited. Considering this is a dataset & benchmark track, the current version does not reach the bar.
10. The paper writing can be improved dramatically (e.g., the readability aspect). No figure is provided. The authors can improve the readability of this paper by adding some figures. it is suggested that the authors DO NOT overuse the BOLD format in both rebuttal and paper.
11. It is not clear enough how the authors ensure the "correctness" of "audited" dataset. Do the authors verify it automatically or manually? It is not explained clearly enough. How to determine "Conflicting Edits"?

[1] MQUAKE: Assessing Knowledge Editing in Language Models via Multi-Hop Questions

[2] Multi-hop Question Answering under Temporal Knowledge Editing

[3] Investigating Multi-Hop Factual Shortcuts in Knowledge Editing of Large Language Models

[4] PokeMQA: Programmable knowledge editing for Multi-hop Question Answering

[5] Cross-Lingual Multi-Hop Knowledge Editing – Benchmarks, Analysis and a Simple Contrastive Learning based Approach

# Questions

What are the new insights obtained from fixing the errors? If no new insights can be obtained, the contribution is limited.

# Compliance with the requirements

It is clearly said that there are four points required as follows. But the authors only provide one single datasheet, and briefly decribe the other three points in the datasheet, which are not clear enough.

It is suggested that the authors use different sections to illustrate the four points separately and thoroughly to make them more clear.

https://neurips.cc/Conferences/2024/CallForDatasetsBenchmarks

> Submission introducing new datasets **must** include the following in the supplementary materials (**as a separate PDF**):
> - **Dataset documentation and intended uses.** Recommended documentation frameworks include datasheets for datasets, dataset nutrition labels, data statements for NLP, data cards, and accountability frameworks.
> - **URL to website/platform where the dataset/benchmark can be viewed and downloaded by the reviewers**
> - **URL to Croissant metadata record documenting the dataset/benchmark available for viewing and downloading by the reviewers.** You can create your Croissant metadata using e.g. the Python library available here: https://github.com/mlcommons/croissant
> - **Author statement that they bear all responsibility in case of violation of rights, etc., and confirmation of the data license.**
Hosting, licensing, and maintenance plan. The choice of hosting platform is yours, as long as you ensure access to the data (possibly through a curated interface) and will provide the necessary maintenance.

**Strengths:**

1.  The authors provide a detailed analysis of various errors in the original dataset, quantifying their impact. For example, they show that up to 76% of cases in MQUAKE-T can be contaminated with just 100 edited cases (Table 1).

2.  MQUAKE-REMASTERED addresses multiple types of errors, including hard corrections for conflicting edits, missing information, and duplicated cases. The authors also introduce dynamic masking to handle contamination issues.

3.   The paper evaluates several open-source knowledge editing methods on the new dataset, providing more reliable performance metrics. This is valuable for the research community to better understand the true capabilities of these methods.

4.  The authors propose a simple yet effective method that outperforms existing baselines on MQUAKE-REMASTERED. This demonstrates that complex, data-specific approaches may not be necessary for good performance.

5.  The authors provide detailed information about their experimental setup, including model versions and computing resources used, which aids in reproducibility.

**Additional Feedback:**

no

**Documentation:**

yes

**Limitations:**

yes

**Opportunities For Improvement:**

# Post-Rebuttal Update on Opportunities For Improvement

I appreciate the rebuttal from the authors.  The following concerns remain. I strongly suggest another round of revision to reach the bar of publication at one top-tier conference.

1. First of all, multi-hop editing is a well-studied field [1-5]. Most of the highly related multi-hop editing benchmarks are not fully discussed in the paper. Since Multi-hop editing is extensively benchmarked in literature, it is strongly suggested that the authors conduct another round of revision to thoroughly discussed the differences compared to all the highly related references [1-5] rather than only [1]. The current version only focuses on discussing the differences with [1], which is not enough for a well-studied field.
2. The motivation is flawed if the authors did not discuss the differences with all the the highly related benchmarks. The authors emphasized too much about one work [1] (e.g., “MQUAKE is flawed due to unintentional clerical and procedural errors “) and ignore the differences with other highly related multi-hop editing benchmarks.
3. The paper structure need to be improved dramatically. The authors use the whole section 2 to explain the MQUAKE benchmark but ignore the other recent multi-hop editing benchmarks. A comprehensive review of the existing benchmarks on multi-hop editing is desired.
4. Related works are highly incomplete and insufficient. Highly related works such as [3,6] are not referenced in this paper. Section 6 should be rewritten to thoroughly discuss the differences with all the highly related references [1-5]. The current version of related work section is no enough.
5. The contribution is limited. The authors only fix the errors in one dataset. No particularly significant insights are concluded.
6. Not much guidance is provided for future multi-hop editing model design. The authors briefly discussed on approach GWalk. No details are provided. It is strongly suggested the authors move the technical content from appendix to the main paper. Page limit is no excuse to put important technical details in the appendix
7. Not much insight can be found. It is strongly suggested that the authors summarize and highlight their insights better. It is hard to find valuable insights from the the current version.
8. The benchmarked methods are highly limited. Based on Table 4, ONLY three existing methods are benchmarked, and most results are OOM. Table 4 actually contains highly limited information.
9. The experiments are highly limited. Considering this is a dataset & benchmark track, the current version does not reach the bar.
10. The paper writing can be improved dramatically (e.g., the readability aspect). No figure is provided. The authors can improve the readability of this paper by adding some figures. it is suggested that the authors DO NOT overuse the BOLD format in both rebuttal and paper.
11. It is not clear enough how the authors ensure the "correctness" of "audited" dataset. Do the authors verify it automatically or manually? It is not explained clearly enough. How to determine "Conflicting Edits"?


[1] MQUAKE: Assessing Knowledge Editing in Language Models via Multi-Hop Questions

[2] Multi-hop Question Answering under Temporal Knowledge Editing

[3] Investigating Multi-Hop Factual Shortcuts in Knowledge Editing of Large Language Models

[4] PokeMQA: Programmable knowledge editing for Multi-hop Question Answering

[5] Cross-Lingual Multi-Hop Knowledge Editing – Benchmarks, Analysis and a Simple Contrastive Learning based Approach

# Compliance with the requirements

It is clearly said that there are four points required as follows. But the authors only provide one single datasheet, and briefly decribe the other three points in the datasheet, which are not clear enough.

It is suggested that the authors use different sections to illustrate the four points separately and thoroughly to make them more clear.

https://neurips.cc/Conferences/2024/CallForDatasetsBenchmarks

> Submission introducing new datasets **must** include the following in the supplementary materials (**as a separate PDF**):
> - **Dataset documentation and intended uses.** Recommended documentation frameworks include datasheets for datasets, dataset nutrition labels, data statements for NLP, data cards, and accountability frameworks.
> - **URL to website/platform where the dataset/benchmark can be viewed and downloaded by the reviewers**
> - **URL to Croissant metadata record documenting the dataset/benchmark available for viewing and downloading by the reviewers.** You can create your Croissant metadata using e.g. the Python library available here: https://github.com/mlcommons/croissant
> - **Author statement that they bear all responsibility in case of violation of rights, etc., and confirmation of the data license.**
Hosting, licensing, and maintenance plan. The choice of hosting platform is yours, as long as you ensure access to the data (possibly through a curated interface) and will provide the necessary maintenance.

**Relation To Prior Work:**

The discussions with related works are highly incomplete and insufficient.

**Summary And Contributions:**

This paper presents a comprehensive audit and fix of the MQUAKE dataset, which is commonly used to evaluate multi-hop knowledge editing methods for large language models. The authors identify several critical issues with MQUAKE, including contamination between edited and unedited cases, missing information in questions, and duplicated cases. They propose MQUAKE-REMASTERED, a fixed version that addresses these issues while maintaining the dataset's capacity. The paper also re-benchmarks existing knowledge editing methods on the new dataset and introduces GWalk, a simple but effective approach to multi-hop knowledge editing.

---

> ### Author Rebuttal · Authors · 2024-08-14
>
> We thank the reviewer for the recognition and detailed feedback. Here, we address your concerns.
>
> ## **[W1 - Some recent methods are not included in the re-benchmarking due to lack of open-source implementations]: As the reviewer recognized, we can only do so much about unreleased methods. But we are working with RAE authors to include it in our camera-ready.**
>
> Our benchmark already featured all open-sourced multiple-hop knowledge editing methods evaluated on MQuAKE (except for DeepEdit, for the slow inference speed reason discussed in `L299`). We hope the reviewer can understand why we can't include unreleased methods, as that would involve significant manpower investment, yet we still won't know if our implementations are faithful.
>
> As a remedy, we will reach out to the authors of these methods and see if we can have their implementation privately and share the benchmark results (though GMeLLo will still be out of reach, as it is an anonymous submission to ACL ARR 2024 Feb, so we don't know who to reach out to).
>
> Further, we recently learned that RAE [7] is accepted, and we are in the process of getting RAE included in our benchmark. We are unsure whether this can happen before August 16 or the rebuttal deadline, as it depends on the bandwidth of the core authors of RAE, but we can promise its inclusion by the time of camera-ready.
>
> ---
>
> ## **[W2 - Evaluate GWalk on another editing dataset]: There isn't another multi-edit multi-hop editing dataset to the best of our knowledge. But here's RippleEdit — a single-edit multi-hop editing dataset.**
>
>
> To the best of our knowledge, there are only two multi-hop knowledge editing datasets with multi-edit designs in mind: MQuAKE, which we already featured, and TKeMQA by Temple-MQA [2]. The latter of which is unreleased.
>
> By "multi-edit," we refer to the scenario where multiple editing facts co-exist at the same time (due to character limitation, we discuss more about the difference between two setups [here](https://openreview.net/forum?id=iTUlYblV0K&noteId=qSbeth2p55)). However, if we relax this to single-edit, we'd have RippleEdit [8], and we can run GWalk on it, so here are the snapshot results on Llama-2-7b-chat:
>
>
> > Single-edit result of RippleEdit-Popular/Recent/Random. C1/2 means the edit is happening at the 1st or the 2nd hop (RippleEdit cases only have 2 hops).
>
>
> | Method | Popular C1 Acc. | Popular C2 Acc. |  Recent C1 Acc. | Recent C2 Acc. |  Random C1 Acc. | Random C2 Acc. |
> |-|-|-|-|-|-|-|
> | ROME | 37.4 | 16.2 |  47.8 | 50.0 | 35.5 | 49.5 |
> | ICE | 85.1 | 67.6 | 74.8 | 85.0 |  73.8 | 80.3 |
> | MeLLo | 45.1 | 77.1 | 50.2 | 80.0 |  40.2 | 68.3 |
> | GWalk (ours) | **85.7** | **81.8** | **80.9** | **87.6** | **76.1** | **82.9** |
>
>
> ---
> We additionally convert RippleEdit to a multi-edit setup — i.e., there are multiple edited facts within the editing knowledge bank at the same time — to a) make it more challenging and, b) show that our audit can also *"fix"* issues within a different dataset. Note we put the "fix" in quotes as RippleEdit is not designed with multi-edit in mind, so the things we fixed are not necessarily errors but just some adjustments required for making a proper multi-edit dataset. In any case, here are the snapshot results on Llama-3-8b-Instruct:
>
>
>
> > Multi-edit result of RippleEdit-Popular/Recent/Random. In this case, we fixed 21/2/0 conflict edits and 3/120/1 case-to-case contamination within RippleEdit-Ropular/Recent/Random datasets, respectively.
>
> | Method | Popular C1 Acc. | Popular C2 Acc. |  Recent C1 Acc. | Recent C2 Acc. |  Random C1 Acc. | Random C2 Acc. |
> |-|-|-|-|-|-|-|
> | MeLLo | 35.1 | 40.3 | 41.1 | 42.4 | 49.5 | 50.0 |
> | GWalk (ours) | **79.0** | **66.9** | **79.2** | **63.9** | **72.9** | **60.0** |
>
>
>
>
>
> ---
>
> ## **[W3 - Lack of human evaluation]: MQuAKE already involves human components as it is Wikidata-based.**
>
> The original MQuAKE and our MQuAKE-Remastered datasets are based on Wikidata. Given the factually rooted nature of asked questions, the automated evaluation is quite reliable.
>
> We are comfortable making this claim because the Wikidata already includes a rich set of verified aliases (likely done by human moderators) for each answer subject. Say if the answer to a question is "UK", any of the following will be correct:
>
> ```
> "answer_alias": [
>           "Britain",
>           "UK",
>           "G. B.",
>           "GBR",
>           "Great Britain",
>           "GB",
>           "G.B.",
>           "United Kingdom of Great Britain and Northern Ireland",
>           "G B",
>           "G B R",
>           "G.B.R.",
>           "\ud83c\uddec\ud83c\udde7",
>           "Great Britain and Northern Ireland",
>           "The UK",
>           "The United Kingdom of Great Britain and Northern Ireland",
>           "U K",
>           "U. K.",
>           "U.K."
>         ]
> ```
>
> So it might be fair to say MQuAKE-based datasets have already included human components for its metrics calculation, making its automatic evaluation more reliable than usual.
>
> ---
>
>
> ## **[W4 - Can the proposed error fix identify issues in other knowledge editing datasets?]: It can in principle, but will need adaptation to the specific data structure. Will add an abstract section for discussion!**
>
>
> Since TKeMQA by Temple-MQA [2] is also Wikidata-based, our auditing tools can surely be applied to TKeMQA should it be released. For other similar dataset — should there be any in the future — our audit analysis would help, but would need to be adopted to the specific data structure of different datasets, as there are many different way to encode the same information. One example is how we "fixed" RippleEdit above in a multi-edit setup.
>
>
> To address the reviewer's interests and to expand our work's impact beyond MQuAKE and multi-hop knowledge editing, we plan to provide a section to conceptualize the type of errors one can typically observe in multi-hop tasks. We hope that can be of guidance for future dataset building in and outside knowledge editing.

---

> ### Author Rebuttal · Authors · 2024-08-14
>
> ## **[W5 - Difference with other multi-hop knowledge editing benchmarks]: None of them is conducted on our MQuAKE-Remastered datasets with errors fixed. Also, their experiment coverage is often way less comprehensive.**
>
> MQuAKE-Remastered is the only dataset that addressed the 5 types of errors that exist in MQuAKE. **Thus, the most significant difference between our MQuAKE-Remastered-based re-benchmark and existing MQuAKE-based evaluations is our benchmark results are obtained on the error-fixed datasets, thus more reliable and reflective.**
>
> Given the major corruption issue of the original MQuAKE dataset, all existing original MQuAKE-based evaluation results are inaccurate and cannot reliably reflect a method's true capability. As we showcased in Table 3:
>
> | Method | CF-3K Original | CF-3K-Remastered | T Original | T-Remastered |
> |-|-|-|-|-|
> | MeLLo | 6.7 | 6.77 | 30.84 | 44.37 |
> | GWalk (ours) | 36.23 | 66.33 | 46.41 | 54.88 |
>
> The performance of the same method may drastically change after proper error fixing.
>
> ---
>
> More so, we'd like to highlight that **our benchmark often significantly beats all reviewer-cited works in terms of model, method, experiment setting, or metrics coverage.** This is mostly because most reviewer-cited papers are not benchmark works but method papers proposing their new editor, where the comprehensiveness felt short of ours due to having a different focus.
>
>
> > Converage comparison among our and reviewer-cited works. For brevity and better relevance, "Method Coverage" only includes open-sourced methods specifically designed for multi-hop editing, as adopted single-hop editors are often too weak to deliver usable results. "Separate Metrics?" means that both the accuracy of edited cases and unedited cases are reported. We consider the inclusion of both metrics paramount, as editing is often a double-edged sword where it will reduce hallucination for some (often edited) questions but induce hallucination for some (often unrelated) questions. Prior work often only test on the former but ignore the latter, we do both.
>
> | Ref. | Dataset Coverage {#edits}| Method Coverage | Separate Metrics? | Error Fix?|
> |-|-|-|-|-|
> | [1] | CF-3k {1, 100, 1000, all}; T {1, 100, 500, all} | MeLLo | No | No |
> | [2] | CF-3K {1, 100, all}; T {1, all} | MeLLo, PokeMQA | No |  No |
> | [3] | CF-3K {all} | N/A  | No | No |
> | [5] | CF-3K {1, 100, all}; T {1, 100, all} | MeLLo, PokeMQA | No | No |
> | **Ours** | CF-3K {1, 100, 1000, all}; T {1, 100, 500, all}; CF-9K {1, 1000, 3000, 6000, all}; CF-6334 {100, 1000, 3000, all} | MeLLo, ICE, IKE, PokeMQA, GWalk, RAE, DeepEdit | Yes | Yes |
>
> *(Our work also often feature more modern mainstream open-sourced models, like Llama-3 and Mistral-v0.2, instead of GPT-J series, which is priminarly used in some of the listed works above.)*
>
> **We highlight that our work, to the best of our knowledge, is the only work that benchmarks on the CF-9K dataset**, which includes 9000+ cases. Not even the original MQuAKE paper has benchmarked on this dataset due to resource limitations. We benchmarked this dataset, and we did it 5 times for all featured methods and with both metrics reported to provide our readers some fine-grained results. We believe our benchmark is miles ahead of all existing MQuAKE evaluations in terms of its fineness and coverage, even if we'd ignore the error-fixing part offered by MQuAKE-Remastered, which we shouldn't.
>
> We additionally note that [4] is not a knowledge editing work, so there is no overlap. [6] focuses on cross-lingual knowledge editing, which is not evaluated on the original MQuAKE and is therefore out of scope (it builds a set of unreleased translated datasets based on MQuAKE, thus also suffering from similar errors and would benefit from our work). It is also appeared on arXiv after the submission deadline (July 14).
>
> ---
>
> We noticed the reviewer adjusted the rating from `5` to `4`, where this W5 is the only modified comment. **We believe our above rebuttal should have addressed concerns in this regard rather comprehensively, but please do let us know if we have interpreted your reservation differently, and we will surely supply a proper rebuttal.**
>
> ---
> [7] Shi et al., Retrieval-Enhanced Knowledge Editing for Multi-Hop Question Answering in Language Models. CIKM 2024.
> [8] Cohen et al., Evaluating the Ripple Effects of Knowledge Editing in Language Models. TACL 2024

---

> ### Comment · Reviewer_wp4u · 2024-08-19
> **I appreciate the rebuttal from the authors. I strongly suggest another round of revision to reach the bar of publication at one top-tier conference.**
>
> # Post-Rebuttal Update
>
> I appreciate the rebuttal from the authors. I strongly suggest another round of revision to reach the bar of publication at one top-tier conference.
>
> 1. First of all, multi-hop editing is a well-studied field [1-5]. Most of the highly related multi-hop editing benchmarks are not fully discussed in the paper. Since Multi-hop editing is extensively benchmarked in literature, it is strongly suggested that the authors conduct another round of revision to thoroughly discussed the differences compared to all the highly related references [1-5] rather than only [1]. The current version only focuses on discussing the differences with [1], which is not enough for a well-studied field.
> 2. The motivation is flawed if the authors did not discuss the differences with all the the highly related benchmarks. The authors emphasized too much about one work [1] (e.g., “MQUAKE is flawed due to unintentional clerical and procedural errors “) and ignore the differences with other highly related multi-hop editing benchmarks.
> 3. The paper structure need to be improved dramatically. The authors use the whole section 2 to explain the MQUAKE benchmark but ignore the other recent multi-hop editing benchmarks. A comprehensive review of the existing benchmarks on multi-hop editing is desired.
> 4. Related works are highly incomplete and insufficient. Highly related works such as [3,6] are not referenced in this paper. Section 6 should be rewritten to thoroughly discuss the differences with all the highly related references [1-5]. The current version of related work section is no enough.
> 5. The contribution is limited. The authors only fix the errors in one dataset. No particularly significant insights are concluded.
> 6. Not much guidance is provided for future multi-hop editing model design. The authors briefly discussed on approach GWalk. No details are provided. It is strongly suggested the authors move the technical content from appendix to the main paper. Page limit is no excuse to put important technical details in the appendix
> 7. Not much insight can be found. It is strongly suggested that the authors summarize and highlight their insights better. It is hard to find valuable insights from the the current version.
>
> [1] MQUAKE: Assessing Knowledge Editing in Language Models via Multi-Hop Questions
>
> [2] Multi-hop Question Answering under Temporal Knowledge Editing
>
> [3] Investigating Multi-Hop Factual Shortcuts in Knowledge Editing of Large Language Models
>
> [4] PokeMQA: Programmable knowledge editing for Multi-hop Question Answering
>
> [5] Cross-Lingual Multi-Hop Knowledge Editing – Benchmarks, Analysis and a Simple Contrastive Learning based Approach

---

> ### Author Response · Authors · 2024-08-19
> **All of your cited works use the original, error-contained, MQuAKE dataset as their main evaluation. There is nothing else to discuss/compare from a dataset/benchmark perspective.**
>
> We appreciate the reviewer for holding our submission to a high standard, but we would like to, again, emphasize two important facts, if they haven't been conveyed clearly.
>
> ## **1. All of your listed works are *NOT* benchmark works. They are method papers proposing their new editors.\* They indeed also evaluate on the original MQuAKE dataset, but their results are not reliable because they didn't fix the errors we audited (see our Table 3), nor is their evaluation coverage comprehensive in comparison to ours (see our [previous response](https://openreview.net/forum?id=iTUlYblV0K&noteId=nxdXMNVPrS)).**
>
> *(\* [2] and [5] also propose their own MQuAKE-based datasets, but they are not publically available.)*
>
> and
>
> ## **2. All of your listed works use the original, error-contained, MQuAKE [1] dataset as their *MAIN* (and often *ONLY*) evaluation. So there is nothing else to discuss or compare from a dataset/benchmark perspective.**
>
>
>
> The reviewer heavily criticizes our lack of discussion of the above reviewer-mentioned "benchmark" works but solely focusing on MQuAKE [1]:
>
> > R`wp4u`: *"Since Multi-hop editing is extensively benchmarked in literature, it is strongly suggested that the authors conduct another round of revision to thoroughly discussed the differences compared to all the highly related references [1-5] rather than only [1]."*
>
> > R`wp4u`: *"The motivation is flawed if the authors did not discuss the differences with all the the highly related benchmarks. The authors emphasized too much about one work [1] ... and ignore the differences with other highly related multi-hop editing benchmarks."*
>
> > R`wp4u`: *"The authors use the whole section 2 to explain the MQUAKE benchmark but ignore the other recent multi-hop editing benchmarks."*
>
> ### **While seemingly unaware of the fact that these "benchmark" works are method works that mainly (or often solely, as of 3 out of 4) evaluated on MQuAKE.**
>
> Here's a breif walkthrough on the dataset coverage of reviewer-mentioned works:
>
> * [1] proposes method MeLLo and the original MQuAKE dataset. **It is only evaluated on MQuAKE.**
> > [1] Section 4: *"We evaluate the following state-of-the-art knowledge editing approaches on our datasets"*
>
> * [2] proposes method Temple-MQA and dataset TKeMQA; neither the dataset nor the method are released. It is **evaluated on MQuAKE, AToKE [7], and TKeMQA. But note TKeMQA is not accessible and AToKe is not multi-hop,** which is the focus of our paper.
> > [2] Section 5.1: *"We evaluate Temple-MQA on a blend of publicly available benchmarks and self-curated datasets. These include: MQUAKE, AToKe, and our newly proposed dataset TKeMQA."*
>
> * [3] is a phenomon analysis paper that **only evaluated on MQuAKE.**
> > [3] Section 3.1: *"Our analysis centers specifically on the MQUAKE-CF-3K dataset released by Zhong et al. (2023)."*
> >
> * [4] proposes method PokeMQA. **It is only evaluated on MQuAKE.**
> > [4] Section 4: *"We evaluate our approach on MQUAKE (Zhong et al., 2023), which is a knowledge editing benchmark."*
>
>
> ### **Thus, there is nothing else to add from a dataset/benchmark perspective after we thoroughly discussed our work's relationship with MQuAKE [1], because that is the same (and often only) dataset everybody runs.**
> Yet, the errors of MQuAKE affects every existing MQuAKE-based evaluation — which includes every work the reviewer mentioned. If anything, the exsitance of these many error-contained evaluations only add to the contribution of our work; because without our audit and fix, scholars will still be referring to these unreliable results.
>
> We hope the reviewer may re-evaluate the initially `5`, previously `4`, and now `3` rating. Because the reviewer is asking for something we already done in essence. We also, once again, emphasize our benchmark is miles ahead of all existing MQuAKE evaluations in terms of its [fineness and coverage](https://openreview.net/forum?id=iTUlYblV0K&noteId=nxdXMNVPrS), even if we'd ignore the error-fixing part offered by MQuAKE-Remastered, which we shouldn't.
>
> ---
>
> ## **3. Reviewer-mentioned [5/6] work is not available by the time of submission (first appeared July 14).**
>
> **For all above conversations, we purposely exclude [5] (Cross-Lingual Multi-Hop Knowledge Editing, which is [6] in the reviewer's original feedback), since:**
>
> (cont. to [next post](https://openreview.net/forum?id=iTUlYblV0K&noteId=0xW9fsD9Lv))
>
> ---
>
> [1] MQUAKE: Assessing Knowledge Editing in Language Models via Multi-Hop Questions
> [2] Multi-hop Question Answering under Temporal Knowledge Editing
> [3] Investigating Multi-Hop Factual Shortcuts in Knowledge Editing of Large Language Models
> [4] PokeMQA: Programmable knowledge editing for Multi-hop Question Answering
> [5/6] Cross-Lingual Multi-Hop Knowledge Editing – Benchmarks, Analysis and a Simple Contrastive Learning based Approach
> [7] History Matters: Temporal Knowledge Editing in Large Language Model

---

> > ### Comment · Reviewer_wp4u · 2024-08-19
> > **The benchmarked methods are highly limited**
> >
> > 8. The benchmarked methods are highly limited. Based on Table 4, ONLY three existing methods are benchmarked, and most results are OOM. Table 4 actually contains highly limited information.
> > 9. The experiments are highly limited. Considering this is a dataset & benchmark track, the current version does not reach the bar.

---

> ### Comment · Reviewer_wp4u · 2024-08-19
> **The relationship between the related works [1-5] and MQUAKE should be better explained in the paper.**
>
> The relationship between the related works [1-5] and MQUAKE should be better explained in the paper. The current version actually does not do this clearly.
>
> [1] MQUAKE: Assessing Knowledge Editing in Language Models via Multi-Hop Questions
>
> [2] Multi-hop Question Answering under Temporal Knowledge Editing
>
> [3] Investigating Multi-Hop Factual Shortcuts in Knowledge Editing of Large Language Models
>
> [4] PokeMQA: Programmable knowledge editing for Multi-hop Question Answering
>
> [5] Cross-Lingual Multi-Hop Knowledge Editing – Benchmarks, Analysis and a Simple Contrastive Learning based Approach

---

> ### Author Response · Authors · 2024-08-19
> **Reviewer-mentioned [5/6] work is not available by the time of NeurIPS D&B deadline. Our GWalk — a simple method — is an addition to a D&B, placing its implementation details in appendix is at least an accpetable pratice.**
>
> ## **3. Reviewer-mentioned [5/6] work is not available by the time of submission (first appeared July 14).**
>
> **For all above conversations, we purposely exclude [5] (Cross-Lingual Multi-Hop Knowledge Editing, which is [6] in the reviewer's original feedback), since:**
>
> * This work presents some cross-lingual datasets built upon MQuAKE, so it is out of scope for general multi-hop KE methods.
> * The proposed cross-lingual datasets are not publically accessible.
> * This paper first appeared on arXiv on **July 14, 2024**, where the NeurIPS 24 D&B has a full paper submission deadline of **June 5, 2024 — there is no way we'd be able to talk about it by the time of submission.**
>
> For these reasons, we respectfully note the following reviewer's criticism is not (fully) proper:
>
> > R`wp4u`: *"Highly related works such as [3,6] are not referenced in this paper. Section 6 should be rewritten to thoroughly discuss the differences with all the highly related references [1-5]. The current version of related work section is no enough."*
>
> **Because [5/6] is unobtainable by the time of submission. [1, 2, 4] are not benchmark works, and we already mentioned them as multi-hop KE *methods* in various places of our Section 2 Preliminary.** We didn't discuss them in Section 6 Related Work because our work is clearly first positioned as a dataset auditing/fixing work, and these works didn't offer any audit or fix to MQuAKE. Further, their evaluation coverages almost don't matter because they are done on the error-contained original MQuAKE dataser, which are unreliable by nature. That being said, we will discuss their coverage in the updated manuscript, which will mostly be integrating the [Coverage Comparision Table](https://openreview.net/forum?id=iTUlYblV0K&noteId=nxdXMNVPrS) we posted in our previous response, which takes minimum effort to add/.
>
> We did miss [3], which is mostly a phenomenon analysis work on multi-hop editing pertaining to the "shortcut" phenomenon, where the LLM associates the final answer of a multi-hop question directly to its initial question as a whole without proper breakdown. We agree it can be a nice addition to mention — which we will do — **but we argue it shouldn't be a fairground for score decrease, especially given** it didn't touch on any KE method specifically designed for multi-hop KE, and **its [code is only avaliable by June 1](https://github.com/Jometeorie/MultiHopShortcuts), which is <5 days to the June 5 submission deadline of NeurIPS D&B.**
>
> ---
>
> ## **4. Auditing/fixing an existing dataset is a well-recognized contribution, as endorsed by Call for Paper. It can potentially provide more impact than making a new one from scratch.**
>
> The reviewer writes:
>
> > R`wp4u`: *"The contribution is limited. The authors only fix the errors in one dataset. No particularly significant insights are concluded."*
>
> **We only fix the errors in one dataset because this is the only multi-edit multi-hop knowledge editing dataset avaliable to the best of our knowledge. Yet almost all exisiting multi-hop knowlegde editing works evaluate on this dataset.**
>
> **We emphasize that auditing/fixing an existing dataset brings a well-recognized contribution to the community, where "audits of existing datasets" is literally written in the [Call for Paper](https://neurips.cc/Conferences/2024/CallForDatasetsBenchmarks)**. We'd even venture to argue that fixing an established dataset can potentially provide more impact than making a new one, as the dataset has already received plenty of community traction, so any error within such dataset — especially in the scale of MQuAKE — already posts a significant issue to the community, which our work shall address.
>
> ---
>
> ## **5. GWalk — a method — is an addition to our contribution, putting its implementation details to appendix of a D&B paper is an acceptable practice.**
>
> > R`wp4u`: *"Not much guidance is provided for future multi-hop editing model design. The authors briefly discussed on approach GWalk. No details are provided. It is strongly suggested the authors move the technical content from appendix to the main paper. Page limit is no excuse to put important technical details in the appendix"*
>
> We again emphasize we are submitting a D&B paper, where the main contribution of our work lies in **auditing and fixing** the error-contained MQuAKE dataset to MQuAKE-remastered and **re-benchmark** almost all available MQuAKE-evaluated multi-hop knowledge editing methods. **It goes without saying that GWalk — a simple *method* we presented — is clearly an addition to a typical D&B paper, so we think it is sane to put its implementation details like pseudocode in the appendix,** as it is even common to see method papers do this.
>
> We further note GWalk is only there to show that it is possible to achieve good performance on MQuAKE-based datasets without leveraging its unique data properties, thus as a guidance of *Making Faithful Approach to ... MQUAKE-Remastered* (see Sec 5.3 for details).

---

> ### Comment · Reviewer_wp4u · 2024-08-19
> **10. No figure is provided. The authors can improve the readability of this paper by adding some figures.**
>
> 10. No figure is provided. The authors can improve the readability of this paper by adding some figures.

---

> > ### Comment · Reviewer_wp4u · 2024-08-19
> > **11. It is not clear enough how the authors ensure the "correctness" of "audited" dataset.**
> >
> > 11. It is not clear enough how the authors ensure the "correctness" of "audited" dataset.

---

> > ### Author Response · Authors · 2024-08-19
> > **"Adding figures" — Sure; "It is not clear enough how the authors ensure the 'correctness' of 'audited' dataset" — We run our aduit again.**
> >
> > We appreciate the reviewer's continuous interest in our work and likely quickly address #10 and #11 post-rebuttal comments mentioned by the reviewer.
> >
> > > R`wp4u`: *"10. No figure is provided. The authors can improve the readability of this paper by adding some figures."*
> >
> > We agree that a figure can be a nice addition, especially to the technical implementation of GWalk in Appendix D.1; we will do so in our updated manuscript. If there is any specific area the reviewer feels a figure would help in, we would love to hear about it.
> >
> > On the same note, we'd venture to argue our paper has decent readability.
> >
> > > R`LCmw:` *"The paper is well written and presents clear background information for a wide audience."*
> >
> > > R`jSuo`: *"Overall, this paper is well-written and well-motivated"*
> >
> > Even for reviewer that criticize our presentation, i.e. R`ujS8`:
> >
> > > R`ujS8`: *"The presentation of this paper has significant room for improvement. (1) The writing style overusing bold texts and section titles makes the paper somehow messy and a little bit hard to follow. (2) The presentation of the experimental results (Table 3 and 4) makes them hard to interpret."*
> >
> > The comment is more about section title and table formatting, instead of figures. We hope the reviewer may find the same as you seem to grasp our four major contributions fairly well in your initial review.
> >
> > ---
> >
> > > R`wp4u`: *"11. It is not clear enough how the authors ensure the "correctness" of "audited" dataset."
> >
> > We run our audit again to make sure our Remasstered dataset doesn't have such type of errors, e.g., there is no contamination stats like Table 1 & 2 in the Remastered dataset.

---

> ### Author Response · Authors · 2024-08-19
> **Our insight is clear as agreed by all other reviewers. We already benchmarked all expect one publically avaliable multi-hop knowledge editing methods evaluated on MQuAKE.**
>
> ## ***6. "Not much insight can be found"* — `The popular MQuAKE datasets contain many errors; we audit them, fixed them, and re-benchmark almost all multi-hop knowledge editing methods` are our main insights. We believe our insights are clear, as agreed by all reviewers.**
>
> The reviewer writes:
>
> > R`wp4u`: *"Not much insight can be found. It is strongly suggested that the authors summarize and highlight their insights better. It is hard to find valuable insights from the the current version."*
>
> Please allow us to requote our abstract and contribution summary around `L82`:
>
> > `Abstract`: "we reveal that **up to 33% or 76% of MQuAKE’s questions and ground truth labels are, in fact, corrupted in various fashions due to some unintentional clerical or procedural oversights.** Our work provides a detailed audit of MQuAKE’s error pattern and a comprehensive fix without sacrificing its dataset capacity. Additionally, we benchmarked almost all proposed MQUAKE-evaluated editing methods on our post-fix dataset, MQuAKE-Remastered."
>
> > `L82`: To pave the way for future advancement of multi-hop knowledge editing, we present our work with the following contributions:
> > 1. A comprehensive audit of MQUAKE
> > 2. Fix/remake MQUAKE to MQUAKE-Remastered
> > 3. Extensively re-benchmark of almost all existing multi-hop knowledge editing methods
> > 4. Guidance for future multi-hop knowledge editing development
>
> We believe our insight is clear, as agreed by other reviewers (you-included).
>
> > R`LCmw`: *"Improving the existing dataset for more correct evaluation of multi-hop reasoning and editing capabilities is an important topic."*
>
> > R`jSuo`: *"This comprehensive error analysis is crucial for the research community, as it highlights specific areas of concern within the dataset that may have gone unnoticed."*
> > *"This benchmarking is invaluable as it provides a clear comparison of how different methods perform ..."*
> > *"The paper offers practical guidelines on how to approach datasets like MQUAKE faithfully..."*
>
> >  R`ujS8`: *"This paper makes a very solid contribution on revisiting the pitfalls hidden in an established benchmark and its evaluation practices..."*
>
> Even the exchanging R`wp4u` have written the following in the initial review:
>
> > * *"The authors provide a detailed analysis of various errors in the original dataset, quantifying their impact."* — **This is our insight/contribution 1:** A comprehensive audit of MQUAKE.
> > * *"MQUAKE-REMASTERED addresses multiple types of errors, including hard corrections for conflicting edits, missing information, and duplicated cases."* — **This is our insight/contribution 2:** Fix/remake MQUAKE to MQUAKE-Remastered
> > * *"The paper evaluates several open-source knowledge editing methods on the new dataset, providing more reliable performance metrics. "* — **This is our insight/contribution 3:** Extensively re-benchmark of almost all existing multi-hop knowledge editing methods.
> > * *"The authors propose a simple yet effective method that outperforms existing baselines on MQUAKE-REMASTERED. This demonstrates that complex, data-specific approaches may not be necessary for good performance."* **This is our insight/contribution 4:** Guidance for future multi-hop knowledge editing development.
>
> **So we hope it is fair to say our delivery is at least relatively clear and easy to digest.**
>
> However, we understand that we might suffer from the authors' bias and hope the reviewer can provide more concrete criticism on why our work lacks insight. We believe our contribution is naturally strong — as we offer the only proper audit/fix to an already popular dataset — and is willing to incorporate relevant presentation adjustments per your advice. Thanks in advanced.
>
> ---
>
> ## **7. *"The benchmarked methods/experiment are highly limited"* — but we already benchmarked all except one publically available multi-hop knowledge editing methods evaluated on MQuAKE.**
>
> As we clearly outlined in **Section 5.1 - Compared Methods.** We already benchmarked all except one publically available mult-hop knowledge editing method specifically designed and evaluated on MQuAKE: Which are MeLLo, PokeMQA, ICE, and IKE; we discard DeepEdit for efficiency reasons discussed around `L300`. We recognize that GMeLLo, GLAME, RAE, Stable-KE, and Temple-MQA are also proper methods, but these methods do not have a publically available implementation by the time of submission, and we think the reviewer would agree that it is common practice to exclude unreproducible works. **We additionally argue our experiment coverage cannot be fairly concluded as "highly limited", as we evaluated on way more and way larger datasets, while doing so with way finer granularity and metrics considered.**
>
> That being said, as discussed in our [Response to W1](https://openreview.net/forum?id=iTUlYblV0K&noteId=BcIZIjn6A4), we are working with RAE authors per their recent acceptance and should be able to provide results in a few days.

---

> ### Comment · Reviewer_wp4u · 2024-08-19
> **What are the new insights obtained from fixing the errors? If no new insights can be obtained, the contribution is limited.**
>
> What are the new insights obtained from fixing the errors? If no new insights can be obtained, the contribution is limited.

---

> ### Author Response · Authors · 2024-08-19
> **Isn't fixing the errors of a popular dataset a major contribution by itself? Plus, we did have additional insights.**
>
> First, we argue auditing/fixing the major errors of a popular dataset is not a limited contribution. As recognized by [Call for Paper](https://neurips.cc/Conferences/2024/CallForDatasetsBenchmarks):
>
> > This track welcomes all work on data-centric machine learning research (DMLR) and open-source libraries and tools that enable or accelerate ML research, covering ML datasets and benchmarks as well as algorithms, tools, methods, and analyses for working with ML data. This includes but is not limited to:
> > * Frameworks for responsible dataset development, **audits of existing datasets, identifying significant problems with existing datasets and their use**
>
> Which is exactly what our paper did.  Further, if the reviewer searches for the keyword "insight" on the same page, the reviewer will find that "insight" is highlighted in three separate key points and is not a universal requirement for all submissions.
>
> That being said, we believe our work offers a healthy amount of insights, in addition to error-fixing:
>
> * First, the type of errors can be found in all multi-edit multi-hop knowledge editing datasets as a category. For example, when we converted RippleEdits to multi-edit, we found similar "mistakes" (though this is using RippleEdit in a different way than the authors intended). We are also confident that MQuAKE-based datasets like TKeMQA or cross-lingual datasets in [5/6] will also suffer similar errors, and their authors can improve their datasets by reviewing our audit analysis. Creators of similar datasets can also use our audits as guidance to verify their datasets are free of such errors before releasing.
> * Second, we are the first work to discover that some of the best-performing MQuAKE-evaluated methods actually leverage some data properties unique to MQuAKE. Pointing this out serves as benign reminder to the field, as otherwise scholars might all try to leverage such dataset-specific properties to push for a better numerical result, deviating their efforts from true and general improvement.
> * Third, our GWalk serves a simple, faithful, minimally invasive but performant approach to MQuAKE-like evaluations. This provides a hackable baseline for future development. Our Table 3 shows methods like MeLLo cannot capture dataset-induced error for not being a reliable retriever in the first place, but GWalk can, making this strong baseline to compare against for improvements tracking.
>
> We hope the reviewer may find them helpful.

---

> > ### Comment · Reviewer_wp4u · 2024-08-19
> > **"fixing error" should lead to new insights. Otherwise, the motivation is not strong enough. More insights on how to design better editing models and new findings on existing methods are desired.**
> >
> > "fixing error" should lead to new insights. Otherwise, the motivation is not strong enough. More insights on how to design better editing models and new findings on existing methods are desired.

---

> > > ### Author Response · Authors · 2024-08-19
> > > **We respectfully argue it is improper to claim the motiviation for auditing/fixing errors on a popular dataset is "not strong enough", but we guess we can agree to disagree. And we think we did provide what you asked.**
> > >
> > > First, we respectfully argue it is improper to claim the motivation for auditing/fixing errors on a popular dataset is *"not strong enough."*
> > >
> > > As the reviewer should be well aware by now via [our previous responses](https://openreview.net/forum?id=iTUlYblV0K&noteId=LnNSLoknNt), almost all of your cited works are mainly, or even solely, evaluated on the original, error-contained, MQuAKE dataset. Thus, their evaluation results are unreliable and do not reflect the true capability of the tested methods. Our work provides the only proper fix to MQuAKE and is a remedy for all reviewer-cited (and many more uncited, or future-conducted) evaluations.
> > >
> > > ## **The fixes of our MQuAKE-Remastered single-handedly makes reliable multi-hop knowledge editing evaluation possible again, and we can't think of why the motivation of provide such fixing is "not strong enough," nor why our contribution is limited.**
> > >
> > > But we guess we can agree to disagree in this regard as this is a rather fundamental difference where none of us can likely persuade another. We call for AC's other reviewers help in reaching this verdict.
> > >
> > > ---
> > >
> > > On the other hand, we believe we offered exactly what the reviewer wanted, a.k.a:
> > >
> > > > R`wp4u`: *"More insights on how to design better editing models"*
> > >
> > > We not only offered insights (making a faithful approach without leveraging unique data properties; preserving edited facts in their triples fashion, leveraging the graph topology they come with; better resistance on retrieving unintended materials and thus less hallucination... `Appendix D.1`), we even offered an end-to-end method that is already tested to have better performance.
> > >
> > > > R`wp4u`: *"new findings on existing methods are desired."*
> > >
> > > * We find that some of the existing methods leverage unique data properties, which is first and potentially only discussed in our work (Section 5.3).
> > > * We find that if the retriever is not good enough, contamination does not matter much (Table 3).
> > >
> > > Additionally, the reviewer previously mentioned most of our featured method in Table 4 are OOM, this is because our work is the only work that evaluated the MQuAKE-CF(-9k) dataset, so the editing facts will make typical in-context learning methods like ICE and IKE OOM. We will add this discussion to our updated manuscript. Though we argue this the fact that they OOM shouldn't be a weakness of our work — we are only faithfully reporting their performance, as a benchmark work should.

---

> > > > ### Comment · Reviewer_wp4u · 2024-08-19
> > > > **One note: it is suggested that the authors DO NOT overuse the BOLD format in both rebuttal and paper, which is kind of annoying.**
> > > >
> > > > One note: it is suggested that the authors DO NOT overuse the BOLD format in both rebuttal and paper, which is kind of annoying.

---

> ### Author Response · Authors · 2024-08-19
> **Noted. We would also hope the reviewer may recognize our rebuttal when it resolve your concerns so that our exchange can be more constructive.**
>
> We believe our rebuttal should have addressed your post-rebuttal concerns #1, 2, 3, 4, 8, 9, and 11 rather thoroughly. We hope the reviewer will recognize our rebuttals to a) provide our work with proper and up-to-date reflection and b) inform us that we can agree a certain concern is addressed and does not require revisiting.
>
> While we apologize for overusing bold texts, part of the reason we are doing so is that you never recognize any of our rebuttals but simply move on to similar or different concerns. While we are grateful for having an engaging reviewer like you and will keep up our end of engagement throughout the way, we would appreciate some proper recognition so that our exchange can be more constructive.

---

> ### Author Response · Authors · 2024-08-21
> **RAE results as promised, as well as additinal DeepEdit results — Our pilot method, GWalk, still outperforms all existing methods significantly.**
>
> Previously, in [*our response to reviewer's W1 - Some recent methods are not included in the re-benchmarking due to lack of open-source implementation*](https://openreview.net/forum?id=iTUlYblV0K&noteId=BcIZIjn6A4), we promised to provide results on RAE, a recently accepted multi-hop knowledge editing method also evaluated on MQuAKE. By working with their core authors, we are glad to provide such results. We additionally present DeepEdit results under the same settings to make our report more comprehensive. **Experiment results suggest GWalk still significantly outperforms all methods.**
>
> > DeepEdit, RAE, and our GWalk on MQuAKE-Remastered-CF-6334 with Llama-3-8B-Instruct, for four different #-edit settings. The following results are obtained under the same setup as Table 12, so they are directly comparable. Results on more models/datasets will be added in our updated manuscript.
> >
> > Header Definition:
> > **#-Edit**: # of cases are considered edited;
> > **Test Edited**: Acc. of edited cases uniquely exist in the test set;
> > **Overlapped Edited**: Acc. of edited cases that exist both in the train and test sets (this metric is unique to the CF-6334 dataset, as this dataset is made with parameter-based methods in mind, where a consistant and constant knowledge base between train and test set is required. More on this in Appendix C.1);
> > **Test Unedited**: Acc. of unedited cases exist in the test set.
> > We additionally note the originally RAE design extracts few-shot examples from {MQuAKE-CF(-9K) - MQuAKE-CF-3K}, we specifically turned off this step due to the low diversity concern between two datasets, as discussed around `L338`, as well as to be aligned with other Table 12-featured methods, like PokeMQA.
>
>
>
>
>
>
> |Method|#-Edit|Test Edited|Overlapped Edited|Test Unedited|Overall Acc.|
> |-|-|-|-|-|-|
> |DeepEdit|100|11.1|19.78|24.29|24.13|
> |RAE|100|22.2|12.09|29.74|29.33|
> |GWalk (ours)|100|**33.3**|**74.73**|**66.92**|**67.01**|
>
> |Method|#-Edit|Test Edited|Overlapped Edited|Test Unedited|Overall Acc.|
> |-|-|-|-|-|-|
> |DeepEdit|1000|8.16|20.52|26.27|24.35|
> |RAE|1000|33.67|11.67|32.49|25.65|
> |GWalk (ours)|1000|**47.45**|**80.94**|**70.65**|**71.89**|
>
> |Method|#-Edit|Test Edited|Overlapped Edited|Test Unedited|Overall Acc.|
> |-|-|-|-|-|-|
> |DeepEdit|3000|7.57|19.65|25.38|21.01|
> |RAE|3000|23.11|10.12|33.48|15.59|
> |GWalk (ours)|3000|**54.05**|**81.6**|**71.12**|**73.76**|
>
> |Method|#-Edit|Test Edited|Overlapped Edited|Test Unedited|Overall Acc.|
> |-|-|-|-|-|-|
> |DeepEdit|6334|7.48|18.81|28.57|18.90|
> |RAE|6334|18.75|11.39|28.57|11.58|
> |GWalk (ours)|6334|**61.02**|**80.47**|**73.02**|**74.22**|
>
> ---
>
> With the two method added, we'd have the following **Experiment Coverage Comparision Table**:
>
> > Converage comparison among our and reviewer-cited works. For brevity and better relevance, "Method Coverage" only includes open-sourced methods specifically designed for multi-hop editing, as adopted single-hop editors are often too weak to deliver usable results. "Separate Metrics?" means that both the accuracy of edited cases and unedited cases are reported.
>
> | Ref. | Dataset Coverage {#edits}| Method Coverage | Separate Metrics? | Error Fix?|
> |-|-|-|-|-|
> | [1] | CF-3k {1, 100, 1000, all}; T {1, 100, 500, all} | MeLLo | No | No |
> | [2] | CF-3K {1, 100, all}; T {1, all} | MeLLo, PokeMQA | No |  No |
> | [3] | CF-3K {all} | N/A  | No | No |
> | [5] | CF-3K {1, 100, all}; T {1, 100, all} | MeLLo, PokeMQA | No | No |
> | **Ours** | CF-3K {1, 100, 1000, all}; T {1, 100, 500, all}; CF-9K {1, 1000, 3000, 6000, all}; CF-6334 {100, 1000, 3000, all} | MeLLo, ICE, IKE, PokeMQA, GWalk, RAE, DeepEdit | Yes | Yes |
>
>
> ---
>
> Recall that two of the reviewer's prior post-rebuttal concerns are:
>
> > *"8. The benchmarked methods are highly limited. Based on Table 4, ONLY three existing methods are benchmarked, and most results are OOM. Table 4 actually contains highly limited information."*
> > *"9. The experiments are highly limited. Considering this is a dataset & benchmark track, the current version does not reach the bar."*
>
> We don't know if the reviewer has found [our previous rebuttal](https://openreview.net/forum?id=iTUlYblV0K&noteId=IumAG2rCHf) in this regard helpful. But for closure, we'd like to additionally note not all methods are suitable to the evaluation used in Table 4 (e.g., PokeMQA). Yet, Table 4 features the biggest MQuAKE-CF-9K dataset nobody else ever evaluated — not even the original MQuAKE creators — so, such OOM reports signify the otherwise unknown shortcomings of some existing methods, which we respectfully argue should be viewed as a plus of our work.
>
> And in any case, we hope the new results on RAE and DeepEdit will statstify the reviewer's interests, as with these, our work presents the most comprehensive benchmark of multi-hop knowledge editing with a healthy margin to the runner-up, yet our GWalk still dominates in the performance department.
>
> ---
>
> RAE:  Retrieval-Enhanced Knowledge ...  in Language Models. CIKM 24

---

### Official Review · Reviewer_jSuo · 2024-07-26
**A good contribution to multi-hop knowledge editing.**

**Rating:** 8
**Confidence:** 4
**Correctness:** Yes
**Clarity:** Yes

**Review:**

This paper reveals that up to 33% to 76% of MQUAKE's questions and ground truth labels are corrupted due to various unintentional clerical or procedural oversights. This work provides a detailed audit of MQUAKE's error patterns and offers a comprehensive fix without sacrificing the dataset's capacity.  The paper goes beyond merely identifying and fixing errors by benchmarking almost all proposed MQUAKE-evaluated editing methods on the improved dataset, MQUAKE-REMASTERED. This benchmarking is invaluable as it provides a clear comparison of how different methods perform on the corrected dataset.

The paper offers practical guidelines on how to approach datasets like MQUAKE faithfully, emphasizing the need for minimally invasive methods that avoid exploiting data-specific properties.


Overall, this paper is well-written and well-motivated. Is it able to provide lifelong knowledge editing results on your MQuAKE-Remastered?

**Strengths:**

This paper offers an in-depth audit of the MQUAKE dataset, identifying that a significant portion of the data, ranging from 33% to 76%, is corrupted due to clerical or procedural oversights. This comprehensive error analysis is crucial for the research community, as it highlights specific areas of concern within the dataset that may have gone unnoticed.

The paper goes beyond merely identifying and fixing errors by benchmarking almost all proposed MQUAKE-evaluated editing methods on the improved dataset, MQUAKE-REMASTERED. This benchmarking is invaluable as it provides a clear comparison of how different methods perform on the corrected dataset.

The paper offers practical guidelines on how to approach datasets like MQUAKE faithfully, emphasizing the need for minimally invasive methods that avoid exploiting data-specific properties.

**Additional Feedback:**

NA

**Documentation:**

Yes

**Limitations:**

Yes

**Opportunities For Improvement:**

Is it able to provide lifelong knowledge editing results on your MQuAKE-Remastered?

Missing some references:

Robust and Scalable Model Editing for Large Language Models

Unveiling the Pitfalls of Knowledge Editing for Large Language Models

Learning to Edit: Aligning LLMs with Knowledge Editing

Editing Large Language Models: Problems, Methods, and Opportunities

**Relation To Prior Work:**

Yes

**Summary And Contributions:**

This paper reveals that up to 33% to 76% of MQUAKE's questions and ground truth labels are corrupted due to various unintentional clerical or procedural oversights. This work provides a detailed audit of MQUAKE's error patterns and offers a comprehensive fix without sacrificing the dataset's capacity. Additionally, the authors benchmarked nearly all proposed MQUAKE-evaluated editing methods on the post-fix dataset, MQUAKE-REMASTERED. The authors observed that many methods attempt to overfit the original MQUAKE by exploiting data-specific properties. The authors provide guidelines on how to approach such datasets faithfully and demonstrate that a simple, minimally invasive approach can achieve excellent editing performance without such exploitation.

---

> ### Author Rebuttal · Authors · 2024-08-14
>
> We thank the reviewer for the much support and recognition. Here, we address your remaining questions and concerns.
>
> ## **[Q1 - Is it able to provide lifelong knowledge editing results on your MQuAKE-Remastered?]: Yes, and we already "accidentally" showed it.**
>
> “Lifelong knowledge editing” often refers to the ability to handle multiple editing operations cumulated over time. In our MQuAKE-Remastered dataset, we offered our dataset with multiple predefined edit-unedit splits (a.k.a. editing intensity), where the number of edited cases grows — e.g., in MQuAKE-Remastered-CF, the dataset is evaluated at `{1, 1000, 3000, 6000, 9171}` edits. So, suppose a method is performant across all these editing intensities; we'd say it is fairly likely that this method is capable of lifelong editing on MQuAKE-Remastered. Similarly, MQuAKE-Reamastered-T is evaluated at `{1, 100, 400, 1864}` edits, and each of its edits is temporal in nature (e.g., "who is the CEO of a certain compnay after a certain time?"), further fitting the lifelong knowledge update scenario. We emphasize that users can evaluate our dataset at even finer editing granularity, should there be any interest.
>
> We additionally note that our proposed method, GWalk, seems like a strong lifelong editor, as it can deliver similar performance across various different editing intensities, as showcased in Tables 4, 10, 11, and 12. Here, we provide a snapshot for the reviewer's convenience:
>
> > Vicuna-7b-v1.5 on MQuAKE-Remastered-CF (Source: Table 4)
>
> | Method | 1-edit | 1000-edit | 3000-edit | 6000-edit | 9171-edit |
> |-|-|-|-|-|-|
> | MeLLo | 22.55 | 21.54 | 17.79 | 12.62 | 6.95 |
> | GWalk (ours) | 61.89 | 56.98 | 56.38 | 53.93 | 54.15 |
>
> It can been seen that GWalk is a lot more consistent compared to MeLLo under a lifelong edit life cycle.
>
> ---
>
> ## **[W1 - Add discussion regarding non-multihop knowledge editing works]: Sure! We will mention them in Section 6 and add a through discussion in Appendix E.**

---

> > ### Comment · Reviewer_jSuo · 2024-08-17
> >
> > Thank you for your response, I think this work makes a clear contribution and I will maintain my score.

---

> > ### Author Response · Authors · 2024-08-17
> > **Thanks again, and a note to all reviewers.**
> >
> > We again thank Reviewer `jSuo` for your recognition and support. D&B has a pretty extended rebuttal window; should there be anything else you'd like to clarify or request, please let us know anytime.
> >
> > To avoid spamming your inbox, we would like to borrow this chance to highlight to all reviewers that we have additionally converted the originally single-edit dataset, RippleEdits [1], to a multi-edit setup like MQuAKE. We then audited/*"fixed"* the multi-edit-converted RippleEdits and presented its result w.r.t. MeLLo and GWalk (ours), where our simple method still prevails ([link to report](https://openreview.net/forum?id=iTUlYblV0K&noteId=qSbeth2p55)). We just want to put this information here since two of our reviewers directly or indirectly touched on this exact dataset, and our single/multi-edit results should have satisfied the relevant interests.
> >
> > ---
> > [1] Cohen et al., Evaluating the Ripple Effects of Knowledge Editing in Language Models. TACL 2024

---

> ### Author Response · Authors · 2024-08-19
> **Reviewer wp4u is visibly hostile and conducted improper score decrease (5 to 4 to 3) upon our fact-based corrections/clarifications. Can you help us? (1/3)**
>
> We like to bring to the reviewer's attention that our [Reviewer `wp4u`](https://openreview.net/forum?id=iTUlYblV0K&noteId=FfFaLjUH0Z), who initially rated our work with a `5`, later decreased such rating to a `4`, and then to a `3`. It goes without saying that making faithful score adjustments — negative or not — during rebuttal is totally proper and is the main purpose of author-reviewer discussion. **We can't help but find that R`wp4u`s score decrease lacks a proper justification, and his/her comments to be hostile. Further, some of R`wp4u`'s comments went against common field practices and showcased a lack of basic discussion etiquette, open-mindedness, or reading comprehension ability.**
>
>
> We know those are big claims, and we are making these accusations extra sparingly.
>
> ---
>
> ## **We also know it is a big ask to request you to review our lengthy exchange with another reviewer. But we don't think it is very easy for our AC to dive into those muddy and potentially developing interactions. You showed much support and a clear understanding of our work, so we'd hope maybe there's something you can do.**
>
> ---
>
> Unfortunately, our exchange with R`wp4u` is very lengthy because R'wp4u repeats the same material in multiple places (like the [initial review](https://openreview.net/forum?id=iTUlYblV0K&noteId=FfFaLjUH0Z)) and enjoys adding new concerns whenever we respond to the prior ones (and doing so without recognition of our prior responses, so we are always unsure whether we'd need to repeat ourselves on a certain issue). But we will do our best to deliver a faithful yet concise picture of what happened.
>
> ## **1. Reviewer `wp4u` can't grasp the difference between evaluating an error-fixed vs an error-contained dataset.**
>
> Initially, R`wp4u` rated our work to be `5`. But before we are able to post a rebuttal, the reviewer modified the feedback with the following additional details and decreased the score to `4`:
>
> > R`wp4u`: *"Multi-hop editing is extensively benchmarked in literature [1-6]. It is unclear what is the unique contribution in this paper. More discussions are desired for the differences and relation with previous literature."*
>
> We think this concern is rather groundless and easy-to-address, as
>
> 1. These works are not benchmark works; they are mostly method works evaluated on MQuAKE. Our experiment coverage is a lot more comprehensive in almost all experiment criteria, and we already featured many of them as methods.
> 2. Our work is the only one that provides a proper error fix to MQuAKE, meaning only our results are reliable and reflect the true capability of a method (and we can show this empirically via experiments, as in Table 3).
>
>
> So we explained exactly that to the reviewer in our [initial rebuttal](https://openreview.net/forum?id=iTUlYblV0K&noteId=nxdXMNVPrS). Highlighting the following:
>
> > `Authors`: "the most significant difference between our MQuAKE-Remastered-based re-benchmark and existing MQuAKE-based evaluations is our benchmark results are obtained on the error-fixed datasets, thus more reliable and reflective."
> > `Authors`: "our benchmark often significantly beats all reviewer-cited works in terms of model, method, experiment setting, or metrics coverage."
>
> And the fact that [4] is not a knowledge editing work, so it is an improper citation.
>
> ---
>
> The reviewer somehow ignored everything we said, decreased our score to `3`, and [again criticized](https://openreview.net/forum?id=iTUlYblV0K&noteId=65zWXBL2Fp) our lack of discussion of the above reviewer-mentioned "benchmark" works but solely focusing on MQuAKE [1]:
>
>
> > R`wp4u`: *"Since Multi-hop editing is extensively benchmarked in literature, it is strongly suggested that the authors conduct another round of revision to thoroughly discussed the differences compared to all the highly related references [1-5] rather than only [1]."*
> > R`wp4u`: *"The motivation is flawed if the authors did not discuss the differences with all the the highly related benchmarks. The authors emphasized too much about one work [1] ... and ignore the differences with other highly related multi-hop editing benchmarks."*
> > R`wp4u`: *"The authors use the whole section 2 to explain the MQUAKE benchmark but ignore the other recent multi-hop editing benchmarks."*
> > R`wp4u`: *"Section 6 should be rewritten to thoroughly discuss the differences with all the highly related references [1-5]"*
>
> *(Honestly, we don't understand what is the major difference among the above four points and why such paraphrases to four concerns are needed, but we digress.)*
>
>
> While seemingly unaware of the fact that these "benchmark" works are method works that mainly or solely (as of 3 out of 4 works) evaluated on MQuAKE, despite our previous rebuttal. Thus, there is nothing else to add from a dataset/benchmark perspective after we thoroughly discussed our work's relationship with MQuAKE [1], because that is the same (and often only) dataset everybody runs.

---

> ### Author Response · Authors · 2024-08-19
> **Can you help us? (2/3)**
>
> ## **Cont 1. Reviewer `wp4u` can't grasp the difference between evaluating an error-fixed vs an error-contained dataset.**
>
>
> We honestly don't understand where the criticism comes from. So we again emphasize the above points, added [a bunch of quotes](https://openreview.net/forum?id=iTUlYblV0K&noteId=LnNSLoknNt) from different papers to support our "those methods are solely evaluated on MQuAKE" claims, and noted one of the reviewer-cited work is out of scope and appeared way later than the submission deadline (deadline: June 5, [6]: July 14). **We think it is almost surreal we have to explain to another NeurIPS reviewer the importance of evaluating on datasets with correct instructions and labels.**
>
> ---
>
> ## **2. Reviewer `wp4u` lacks the ability to digest basic information from ML literature/rebuttal responses, and criticizes us for not doing the impossible.**
>
> The reviewer again ignored everything we said, [repeat the comparison argument](https://openreview.net/forum?id=iTUlYblV0K&noteId=o4Ez4zRum8) (**including [6], after we specifically pointed out this exact work is not available by the time of submission**), and move on to [saying](https://openreview.net/forum?id=iTUlYblV0K&noteId=Jc6xozGVSE) our benchmark is limited because we featured too few methods and showing undesired method performance.
>
> > R`wp4u`: *"The benchmarked methods are highly limited. Based on Table 4, ONLY three existing methods are benchmarked, and most results are OOM. Table 4 actually contains highly limited information."*
>
> While ignoring the fact not all methods are suitable in Table 4's evaluation, as we discussed in our paper (Sec 4.2). Also, we have already featured all except one open-sourced methods specific to MQuAKE. **It is common practice to exclude non-open-sourced methods for benchmark work, but the reviewer seemingly disagrees.**
>
> More so, we think it is absurd to criticize us for a method's Out-Of-Memory report. As we [previously noted](https://openreview.net/forum?id=iTUlYblV0K&noteId=nxdXMNVPrS) to the reviewer, Table 4 features the biggest MQuAKE-CF-9K dataset nobody else ever evaluated — not even the original MQuAKE creators — so if anything, such OOM reports signify the otherwise unknown shortcomings of some existing methods, which should be viewed as a plus. But again, the reviewer seemingly disagrees.
>
> ---
>
>
> Last, the reviewer claims our experiments are limited, but without specifying any reason.
>
> > R`wp4u`: *"The experiments are highly limited. Considering this is a dataset & benchmark track, the current version does not reach the bar."*
>
> Despite the fact that our experiment coverage exceeds all existing works for quite a margin. And we have already provided a comprehensive [experiment coverage comparison table](https://openreview.net/forum?id=iTUlYblV0K&noteId=nxdXMNVPrS) to this reviewer.
>
> **So we really struggle to understand how the reviewer can reach the conclusion of our experiment is *"highly limited"* in lieu of ... simple facts (which we have summed up and provided right under the reviewer's nose).** The only conclusion we can unfortunately reach is the reviewer lacks common field understanding and the ability to digest basic information — like dataset/experiment coverage, availability of a certain method, or the date a paper first appears — from typical ML literature or a few hundred words of rebuttal texts, and we are the victim.
>
>
> ---
>
> [1] MQUAKE: Assessing Knowledge Editing in Language Models via Multi-Hop Questions
> [2] Multi-hop Question Answering under Temporal Knowledge Editing
> [3] Investigating Multi-Hop Factual Shortcuts in Knowledge Editing of Large Language Models
> [4] MoreHopQA: More Than Multi-hop Reasoning
> [5] PokeMQA: Programmable knowledge editing for Multi-hop Question Answering
> [6] Cross-Lingual Multi-Hop Knowledge Editing – Benchmarks, Analysis and a Simple Contrastive Learning based Approach

---

> ### Author Response · Authors · 2024-08-19
> **Can you help us? (3/3)**
>
> ## **3. Reviewer `wp4u` thinks the motivation for auditing/fixing errors on a popular dataset is "not strong enough," and our contribution is "limited."**
>
>
> **We think these are some borderline nonsensical hot takes.**
>
> The reviewer repeatedly cited multiple works for us to compare, all evaluated on the same error-contained MQuAKE datasets, all producing and relying on unreliable results. Our work fixes such exact datasets and single-handedly makes reliable multi-hop knowledge editing evaluation possible again, yet somehow its motivation is *"not strong enough"* and its contribution is *"limited"*? We can't follow this logic.
>
> We mentioned this contribution is well recognized by the [Call for Paper](https://neurips.cc/Conferences/2024/CallForDatasetsBenchmarks):
>
> > This track welcomes all work on data-centric machine learning research (DMLR) and open-source libraries and tools that enable or accelerate ML research, covering ML datasets and benchmarks as well as algorithms, tools, methods, and analyses for working with ML data. This includes but is not limited to:
> > * Frameworks for responsible dataset development, **audits of existing datasets, identifying significant problems with existing datasets and their use**
>
> And argue that fixing an established dataset can potentially provide more impact than making a new one, as the dataset has already received plenty of community traction, so any error within such dataset — especially at the scale of MQuAKE — already posts a significant issue to the community, which our work shall address. But the reviewer believes he/she knows better and commented:
>
>
> > R`wp4u`: *"What are the new insights obtained from fixing the errors? If no new insights can be obtained, the contribution is limited"*
> > R`wp4u`: *"'fixing error' should lead to new insights. Otherwise, the motivation is not strong enough. More insights on how to design better editing models and new findings on existing methods are desired."*
>
> **We believe these comments showcased the lack of basic open-mindedness to appreciate works that contribute in a manner outside the reviewer's personal preference.** Not to mention that our work actually does provide such insights — e.g., we literally implemented and evaluated an end-to-end method, GWalk, that beats SOTA by up to 53.84%. What adds more to *"how to design better editing models"* than making one that works and showing how it works?
>
> ---
> ## **4. Reviewer `wp4u` lacks basic discussion etiquette for always adding more "concerns" on the fly whenever a prior concern is responded to; but never confirms whether our prior rebuttals have resolved any of the raised concerns. And is mad about us because we emphasize on issues.**
>
> The reviewer always adds more concerns whenever we respond to the existing ones. While we appreciate the reviewer for being engaging, **the fact that the reviewer never confirms whether our prior rebuttals have resolved any of the raised concerns, and our observation that those newly added concerns are increasingly boilerplate and cosmetic by nature** (along the lines of *"move appendix material to main text"*, *"benchmark/experiment are limited"*, *"add figures"*, etc.) **makes us speculate the reviewer is not engaging to help us improve our work, but rather trying to win an ego battle,** potentially because we respectfully pointed out some errors in the reviewer's initial reviews.
>
> (The reviewer [later took issue](https://openreview.net/forum?id=iTUlYblV0K&noteId=N3wRYLBUnt) with our overuse of bold texts, which we first admit we might have indeed used rather excessively, and we apologize. But the fact that the reviewer repeated three times on the same error-fixed vs error-contained issue ([#1](https://openreview.net/forum?id=iTUlYblV0K&noteId=FfFaLjUH0Z), [#2](https://openreview.net/forum?id=iTUlYblV0K&noteId=65zWXBL2Fp), and [#3](https://openreview.net/forum?id=iTUlYblV0K&noteId=o4Ez4zRum8)) while ignoring our multiple rebuttals, make us have no choice but to emphasize some basic talking points over and over.)
>
> We don't think this is a constructive way to communicate, though we have no choice but to engage in this hostile, and potentially endless, exchange, purposely crafted by the reviewer.
>
> ---
>
> We apologize for the long read and the rather scorching expressions. But we hope our writing will help highlight the unprofessional score decreasing of Reviewer `wp4u` and, maybe via your intervention as another reviewer, may earn us a proper final evaluation on our work. Again, we sincerely appreciate your assistance in ensuring an unbiased review of our work and would wholeheartedly respect your final decision, wether you decide to engage on this already extra issue or not.
>
> Sincerely,
>
> *Paper840 Authors*

---

> ### Author Response · Authors · 2024-08-21
> **Addtional results on DeepEdit and RAE.**
>
> As highlighted around `L289 - Compared Methods` paragraph, our work featured almost all (-1) reproducible multi-hop knowledge editing methods evaluated on MQuAKE. Unfortunately, there are many more methods we cannot feature due to a lack of open-source implementations, potentially due to most of these methods are still under submission.
>
> We are glad to work with RAE — a recently accepted multi-hop editing method to CIKM, which was also evaluated on MQuAKE — core authors and provided the following pilot results, in addition to our recently obtained DeepEdit reports. **The reviewer didn't ask for any of it, but we thought it might be a nice thing to add — as not only the following results further strengthen the significant performance advantage of our GWalk method, but it also makes our work the [most comprehensive evaluation](https://openreview.net/forum?id=iTUlYblV0K&noteId=TvSHM0uGhA) of multi-hop knowledge editing with a healthy margin over the runner-up.**
>
>
> > DeepEdit, RAE, and our GWalk on MQuAKE-Remastered-CF-6334 with Llama-3-8B-Instruct, for four different #-edit settings. The following results are obtained under the same setup as Table 12, so they are directly comparable. Results on more models/datasets will be added in our updated manuscript.
> >
> > Header Definition:
> > **#-Edit**: # of cases are considered edited;
> > **Test Edited**: Acc. of edited cases uniquely exist in the test set;
> > **Overlapped Edited**: Acc. of edited cases that exist both in the train and test sets (this metric is unique to the CF-6334 dataset, as this dataset is made with parameter-based methods in mind, where a consistant and constant knowledge base between train and test set is required. More on this in Appendix C.1);
> > **Test Unedited**: Acc. of unedited cases exist in the test set.
> > We additionally note the originally RAE design extracts few-shot examples from {MQuAKE-CF(-9K) - MQuAKE-CF-3K}, we specifically turned off this step due to the low diversity concern between two datasets, as discussed around `L338`, as well as to be aligned with other Table 12-featured methods, like PokeMQA.
>
>
>
>
>
> |Method|#-Edit|Test Edited|Overlapped Edited|Test Unedited|Overall Acc.|
> |-|-|-|-|-|-|
> |DeepEdit|100|11.1|19.78|24.29|24.13|
> |RAE|100|22.2|12.09|29.74|29.33|
> |GWalk (ours)|100|**33.3**|**74.73**|**66.92**|**67.01**|
>
>
>
>
>
>
> |Method|#-Edit|Test Edited|Overlapped Edited|Test Unedited|Overall Acc.|
> |-|-|-|-|-|-|
> |DeepEdit|1000|8.16|20.52|26.27|24.35|
> |RAE|1000|33.67|11.67|32.49|25.65|
> |GWalk (ours)|1000|**47.45**|**80.94**|**70.65**|**71.89**|
>
>
>
>
>
> |Method|#-Edit|Test Edited|Overlapped Edited|Test Unedited|Overall Acc.|
> |-|-|-|-|-|-|
> |DeepEdit|3000|7.57|19.65|25.38|21.01|
> |RAE|3000|23.11|10.12|33.48|15.59|
> |GWalk (ours)|3000|**54.05**|**81.6**|**71.12**|**73.76**|
>
>
> |Method|#-Edit|Test Edited|Overlapped Edited|Test Unedited|Overall Acc.|
> |-|-|-|-|-|-|
> |DeepEdit|6334|7.48|18.81|28.57|18.90|
> |RAE|6334|18.75|11.39|28.57|11.58|
> |GWalk (ours)|6334|**61.02**|**80.47**|**73.02**|**74.22**|
>
>
> Cheers,
> *Paper840 Authors*
>
> ---
> DeepEdit: DeepEdit: Knowledge Editing as Decoding with Constraints. arXiv 2024
> RAE: Shi et al., Retrieval-Enhanced Knowledge Editing for Multi-Hop Question Answering in Language Models. CIKM 2024

---

### Official Review · Reviewer_LCmw · 2024-07-26
**Important topic, however, the contribution appears marginal.**

**Rating:** 4
**Confidence:** 4
**Correctness:** Please refer to the above.
**Clarity:** Please refer to the above.

**Review:**

- The work is built upon an existing dataset. While it provided an improved quantitative analysis of the scale of the dataset's shortcomings, the issues have been pointed out in other works and lack novelty / original contributions.
- With respect to the API developed for dynamic masking potential conflicts, does it mean that the total number of edited triples is reduced in each single evaluation round by masking out those triples that are subject to contamination risks? If so, it seems that the overall effect is similar to deleting those triples in each round. Is it better to leverage your tools and remove those conflicting examples, and then "refill" by new conflict-free examples a better strategy to improve the quality of the dataset?
- The proposed editing method is still RAG-based. While it provides a cleaner relational data pool, the method presents a marginal improvement of prior work and lacks clear novelty.

**Strengths:**

- The paper is well written and presents clear background information for a wide audience.
- Improving the existing dataset for more correct evaluation of multi-hop reasoning and editing capabilities is an important topic.

**Additional Feedback:**

N/A

**Documentation:**

github link cannot be opened

**Limitations:**

Please refer to the above.

**Opportunities For Improvement:**

Please refer to the above.

**Relation To Prior Work:**

Prior works have pointed out the flaws in the MQuAKE dataset and proposed certain remedies. This work provided a quantified analysis of the scale of the flaws and provided marginal improvement on the remedy.

**Summary And Contributions:**

This paper intends to address the flaws within the MQuAKE dataset. Improving the quality of the dataset for more correct evaluation of multi-hop reasoning and editing capabilities is an important topic. While the authors provide a more comprehensive and quantitative analysis of the scale of flaws within the dataset, key contamination categories have been pointed out and discussed in prior works, and therefore, the contributions of this particular paper do not appear to be significant enough. Re-evaluation of existing methods is commendable. The proposed improvement on multi-hop editing method discussed appears to lack novelty.

---

> ### Author Rebuttal · Authors · 2024-08-14
>
> We thank the reviewer for the detailed feedback.
>
> ## **[W1 - Limited novelty/contribution as "the issues have been pointed out in other works"]: Not really. Previous works figured at most 1.5 out of the 5 types of errors we identified, offered no taxonomy, no quantitative analysis, and no proper fix. We did all.**
>
>
> In the most respectful way possible, we firmly disagree with this assessment. **Only 1.5 out of 5 errors have been pointed out in previous works, and none of the errors have been properly categorized, measured, or fixed until our work.**
>
>
> We understand that editing tasks are intricate, and some of our related work discussions are left in the Appendix, so please allow us to provide a detailed walkthrough here. Our work identified 5 types of errors in the original MQuAKE dataset, which are:
>
> 1. Intra Contamination (Sec 3.1)
> 2. Inner Contamination (3.2)
> 3. Conflicting Edits (3.3)
> 4. Missing Information in Question Instructions (3.4)
> 5. Duplicated Cases (3.5)
>
> As we thoroughly discussed in Related Works (Sec 6) and Appendix E. GMeLLo [1] briefly touched on the same type of error we discussed in Sec 3.4 (Missing Information in Question Instructions), where they merely showed two error examples, and that's it. **GMeLLo does not provide a quantified measurement of its scale nor any fix. We did both in Sec 3.4 and 4.1.**
>
>
> DeepEdit [2] *partially* discovered the same error we discussed in Sec 3.2 (Inner Contamination). DeepEdit does provide a quantified measurement of its scale but only pertains to the MQuAKE-CF-3K dataset (which is 1/3 of MQuAKE offering), and such quantifiable results are only valid when all cases of MQuAKE-CF-3K are considered edited (which is 1/4 of MQuAKE-CF-3K standard reporting setup). **So, we'd say this is, at best, countable as "discovering 0.5 error," and we are being generous.**
>
> Further, DeepEdit provides a rather hardcore fix to this problem by removing the 998 inner contaminated cases from the MQuAKE-CF-3K dataset. While this fix is, of course, helpful, **our post-fix datasets are much more comprehensive and effective** since they patched many more errors revealed in Sec 3 (which still exists in the DeepEdit-fixed dataset), work outside the MQuAKE-CF-3K dataset, and do not require all cases to be considered edited. **Most importantly, our fix is done so without sacrificing almost 1/3 of the capacity of the original dataset.**
>
> Last, **we respectfully argue for datasets, being the first to identify a particular error is not as important as figuring out the pattern and scale of such errors and providing a proper fix.** E.g., Any of us could spend some time reviewing ImageNet and discover some mislabeled samples ([here are some examples](https://www.researchgate.net/figure/Some-of-the-found-mislabeled-images-in-the-ImageNet-dataset-The-classes-of-the-images_fig3_321400551)). But we believe reviewer would agree that simply finding a few nosisy samples doesn't offer much value until one can put them into a reasonable taxonomy, measure their scales, and provide a proper fix. Our work provided all three to MQuAKE, and we respectfully argue our contribution is therefore strong.
>
> ---
>
> ## **[W2 - Can you refill the masked data points?]: Yes, but there is minimum gain. We are talking about refilling +4 edited facts when there are already thousands to retrieve.**
>
> *(We rethought this issue and now provide an updated response.)*
>
> As we have already removed directly conflicting edits observed in Sec 3.3, there can be at most one contamination per subquestion. Given one MQuAKE multi-hop question has a maximum hop count of 4, we can refill at most 4 edited facts per a contaminated case. We opt not to do it for two reasons: 1) +4 facts provide a marginal increase in hardness, as often there are already thousands of editing facts, and 2) this refill will need to be dynamic, too; unless they are just garbage editing facts that interact with no question.
>
> However, we do realize the importance of having a constant, non-masking dataset, as this is a must-have for parameter-based knowledge editing methods. So **we filtered out 6334 cases that are completely conflict-free from each other (regardless of edit-unedited splits) and present MQuAKE-Remastered-CF-6334.** We discussed it more at `L281` and Appendix C.1, and benchmarked it in Table 12.
>
>
> ---
>
> ## **[W3 - The proposed editing method provides "marginal improvement of prior work and lacks clear novelty"]: Up to 53.84% improvement over PokeMQA and 47.2% over MeLLo is not marginal. Novelty might be slim, but this is D&B track.**
>
>
> First, our GWalk method offers **up to 53.84%** improvement over PokeMQA — the best reproducible method tested on MQuAKE as of submission — in Table 12. And offers **up to 47.2%** improvement over MeLLo (Table 4). **Give the upper bound is only 100%, this kind of drastic performance improvement is absolutely not marginal, by definition.**
>
> In terms of novelty, we agree that storing triplet-rooted data in a graph format is rather intuitive. As discussed in Sec 5.3, we propose the GWalk method for the sole purpose of showing one can achieve good performance without leveraging specific data properties that are unique to MQuAKE datasets. It goes without saying that **proposing a new method is already an addition to dataset & benchmark track papers, so we hope the reviewer will not punish us because we go above and beyond to ensure our users would approach our remastered dataset faithfully, with a good & efficient baseline in hands to track their improvement.** (Note, without a good enough baseline, the Remastering has limited effects — e.g., if a RAG-based method is not able to effectively retrieve the right editing fact, whether the facts are contiminated do not matter much, as showcased in Table 3).
>
>
> ---
> [1] Graph Memory-based Editing for Large Language Models. Submission ACL ARR 2024 Feb
> [2] Wang et al., DeepEdit: Knowledge Editing as Decoding with Constraints. arXiv 2024

---

> > ### Author Rebuttal · Authors · 2024-08-15
> >
> > ## **[W2 Follow-up]:  Additional statistics regarding edited facts refill — the at most +4 refill is indeed a drop in the ocean.**
> >
> > Previously, in our response to W2, we discussed that refilling masked-out edited facts provides *"a marginal increase in hardness, as often there are already thousands of editing facts."* Here, we first confirm that for every case, there are indeed at most 4 edited facts to be masked; as MQuAKE multi-hop questions a maximum of 4 hops, where each hop can be contaminated by at most one edited fact.
> >
> >
> > > MQuAKE-Remastered-CF: Different number of edited cases vs How many cases would require 0/1/2/3/4/5 masks. For example, when all cases are considered edited, 1745 cases would be required to mask off 1 edited fact to be contamination-free.
> >
> >
> > | Number of Edited Cases / Number of Cases that require #-mask  | 0-mask | 1-mask | 2-mask | 3-mask | 4-mask| 5-mask |
> > |-|-|-|-|-|-|-|
> > | All | 6349 | 1745 |  870 |  254 |   0 | 0 |
> > | 6000| 4717 | 2404 | 1350 |  510 |  237 | 0 |
> > | 3000| 3459 | 3189 | 1604 |  731 |  235 | 0 |
> > | 1000| 2999 | 4257 | 1491 |  401 |   70 | 0|
> > | 100 | 4606 | 3526 |  994 |   90 |    2 | 0 |
> >
> >
> > Then, we subject this "at most 4 edited facts to refill" to the size of their editing knowledge bank (how many unique edited facts) and find the following:
> >
> > > MQuAKE-Remastered-CF: Size of editing knowledge bank vs Different numbers of edited cases. For example, when 3000 cases are considered edited, there are 2991 unique edited facts in the editing knowledge bank, making +4 a marginal addition.
> >
> > | Number of Edited Cases | 100 | 1000 | 3000 | 6000 | All (9171) |
> > |-|-|-|-|-|-|
> > | Number of Unique Edited Facts | 150 | 1171 | 2991 | 5137 | 7252 |
> >
> >
> > We believe we can safely conclude that adding at most 4 more edited facts to an editing knowledge bank of the above sizes brings a marginal increase in the problem hardness, and we hope our reviewer agrees.

---

> ### Author Response · Authors · 2024-08-21
> **Additional results on RAE and DeepEdit. Our pilot method, GWalk, still outperforms all existing methods significantly.**
>
> One of your major concerns is our proposed pilot method, GWalk,
>
> > *"presents a marginal improvement of prior work"*
>
> We [previously argued](https://openreview.net/forum?id=iTUlYblV0K&noteId=inLNheuU4T) our GWalk offers non-marginal improvement over existing methods, e.g., Up to +53.84% over PokeMQA.
>
> Here, we further compare GWalk against DeepEdit [2]: the only unfeatured open-sourced multi-hop knowledge editing (MHKE) method available around the time of submission; and RAE [3]: a recently accepted MHKE method to CIKM. **We are glad to report that GWalk still significantly outperforms all methods.**
>
> > DeepEdit [2], RAE [3], and our GWalk on MQuAKE-Remastered-CF-6334 with Llama-3-8B-Instruct, for four different #-edit settings. The following results are obtained under the same setup as Table 12, so they are directly comparable. Results on more models/datasets will be added in our updated manuscript.
> >
> > Header Definition:
> > **#-Edit**: # of cases are considered edited;
> > **Test Edited**: Acc. of edited cases uniquely exist in the test set;
> > **Overlapped Edited**: Acc. of edited cases that exist both in the train and test sets (this metric is unique to the CF-6334 dataset, as this dataset is made with parameter-based methods in mind, where a consistant and constant knowledge base between train and test set is required. More on this in Appendix C.1);
> > **Test Unedited**: Acc. of unedited cases exist in the test set.
> > We additionally note the originally RAE design extracts few-shot examples from {MQuAKE-CF(-9K) - MQuAKE-CF-3K}, we specifically turned off this step due to the low diversity concern between two datasets, as discussed around `L338`, as well as to be aligned with other Table 12-featured methods, like PokeMQA.
>
>
>
>
>
>
> |Method|#-Edit|Test Edited|Overlapped Edited|Test Unedited|Overall Acc.|
> |-|-|-|-|-|-|
> |DeepEdit|100|11.1|19.78|24.29|24.13|
> |RAE|100|22.2|12.09|29.74|29.33|
> |GWalk (ours)|100|**33.3**|**74.73**|**66.92**|**67.01**|
>
>
>
>
>
>
> |Method|#-Edit|Test Edited|Overlapped Edited|Test Unedited|Overall Acc.|
> |-|-|-|-|-|-|
> |DeepEdit|1000|8.16|20.52|26.27|24.35|
> |RAE|1000|33.67|11.67|32.49|25.65|
> |GWalk (ours)|1000|**47.45**|**80.94**|**70.65**|**71.89**|
>
>
>
>
>
> |Method|#-Edit|Test Edited|Overlapped Edited|Test Unedited|Overall Acc.|
> |-|-|-|-|-|-|
> |DeepEdit|3000|7.57|19.65|25.38|21.01|
> |RAE|3000|23.11|10.12|33.48|15.59|
> |GWalk (ours)|3000|**54.05**|**81.6**|**71.12**|**73.76**|
>
>
> |Method|#-Edit|Test Edited|Overlapped Edited|Test Unedited|Overall Acc.|
> |-|-|-|-|-|-|
> |DeepEdit|6334|7.48|18.81|28.57|18.90|
> |RAE|6334|18.75|11.39|28.57|11.58|
> |GWalk (ours)|6334|**61.02**|**80.47**|**73.02**|**74.22**|
>
> We hope this result would clear up the reviewer's concern regarding whether the performance improvement offered by GWalk is marginal or not.
>
> ---
>
> The reviewer two others concerns are our GWalk method
>
> >  *"lacks clear novelty"*
>
> As we have [previously discussed](https://openreview.net/forum?id=iTUlYblV0K&noteId=inLNheuU4T), we have no problem in admitting that the technical novelty of GWalk is slim. In fact, this is the exact reason we present GWalk in a D&B, instead of a method paper of its own. We believe the reviewer would agree that proposing a new method is already consider extra for a D&B paper, **so we hope the reviewer would not grill us too hard because we go above and beyond to ensure our users would approach our remastered dataset faithfully, with a good & efficient baseline in hands to track their improvement.**
>
> and
>
> > *"Prior works have pointed out the flaws in the MQuAKE dataset and proposed certain remedies"*
>
> Again, as we have [previously showcased](https://openreview.net/forum?id=iTUlYblV0K&noteId=inLNheuU4T), prior works only pointed out at most 1.5 out of 5 errors we audited. Only one prior work, DeepEdit [2], attempted in offering a remedy. The DeepEdit-fix only addresses the MQuAKE-CF-3K dataset (which is 1/3 of the MQuAKE offering), can address one type of error when all cases of MQuAKE-CF-3K are considered edited (which is 1/4 of MQuAKE-CF-3K standard reporting setup). The post-DeepEdit-fix dataset has sacrificed almost 1/3 of the original dataset capacity  (998 out of 3000 cases) , yet it still contains other type of errors.
>
> **Our Remastered fix works on all datasets, with no limitation on edit intensity, and do not sacrifice such many cases. We believe it is fair to conclude our remedy is much more superior and comprehensive, and we hope our reviewer would agree.**
>
> ---
> [2] Wang et al., DeepEdit: Knowledge Editing as Decoding with Constraints. arXiv 2024
> [3] Shi et al., Retrieval-Enhanced Knowledge Editing for Multi-Hop Question Answering in Language Models. CIKM 2024

---

> ### Author Response · Authors · 2024-08-23
> **An invitation to discuss as your concerns are rather clear-cut.**
>
> Dear Reviewer `LCmw`,
>
> We believe it is fair to say your initial concerns of our work are rather clear-cut, as they can be concisely summarized in your own words:
>
> > R`LCmw`: *"Prior works have pointed out the flaws in the MQuAKE dataset and proposed certain remedies. This work provided a quantified analysis of the scale of the flaws and provided marginal improvement on the remedy."*
>
> In our previous rebuttals, we walked through why we respectfully disagree with this assessment, since, as far as we know, **only 1.5 out of 5 errors have been pointed out in previous works, and none of the errors have been properly categorized, measured, or fixed until our work.** We believe much of this misunderstanding is due to the intricate nature of editing tasks and how we listed relevant information in both Related Works (Sec 6) and Appendix E.
>
> ## **Here, we'd like to be respectful your time and present this one-stop comparison table:**
>
> > Error Analysis/Quantification/Fix of MQuAKE provided in different works.
>
> | Ref. | Types of Errors Found | Error Quantified | Scope of Error Fixed | Types of Errors Addressed | Cost of Fixing |
> |-|-|-|-|-|-|
> | [GMeLLo](https://openreview.net/forum?id=W2GDC0NiVou) | Missing Instruction | No  | No | No | N/A |
> | [DeepEdit](https://arxiv.org/abs/2401.10471) | Inner Contamination  | MQuAKE-CF-3K in `3000`-edit | MQuAKE-CF-3K in `3000`-edit |  Inner Contamination (4 other types of error still exist post-fixing) | 998 out of 3000 cases removed |
> | Ours | Intra Contamination, Inner Contamination, Conflicting Edits, Missing Instructions, Duplicated Cases | MQuAKE-CF-3K in `{1, 100, 1000, 2000, 3000}`-edit, MQuAKE-CF-T in `{1, 100, 500, 1868}`-edit,  MQuAKE-CF(-9K) in `{1, 100, 1000, 2000, 3000, 5000, 9218}`-edit | MQuAKE-CF-3K in `any`-edit, MQuAKE-CF-T in `any`-edit, MQuAKE-CF(-9K) in `any`-edit. Additionally, we offer MQuAKE-Remastered-CF-6334 in `any`-edit | Intra Contamination, Inner Contamination, Conflicting Edits, Missing Instructions, Duplicated Cases | No case removed |
>
> ## **We are confident this comparison table clearly showcases our work offers non-marginal contribution compared to the few (2) prior arts.**
>
> And we sincerely hope our reviewer may agree with this analysis and provide us a more reflective rating, or let us know what are the remaining concerns. Thanks in advance.

---

> ### Author Response · Authors · 2024-08-28
> **Another reminder if we may, as we are roughly 3-day till the deadline.**
>
> Dear Reviewer `LCmw`
>
> As our previous comments have gone through, your concern of our work can be concisely summarized by your own words:
>
> > R`LCmw`: *"Prior works have pointed out the flaws in the MQuAKE dataset and proposed certain remedies. This work provided a quantified analysis of the scale of the flaws and provided marginal improvement on the remedy."*
>
> We believe our previous [error analysis comparison table](https://openreview.net/forum?id=iTUlYblV0K&noteId=Fo1UdjXdsf), as well as our Related Works (Sec 6) and Appendix E, should have showcased that **our work has figured out many more errors (5, vs 1.5) in comparison to the few (2) prior arts** that also explored the flaws of MQuAKE. Yet, **our fix is much more comprehensive, thorough, and usable than the only available fix provided in one other literature**, as our fix addresses all 5 types of mistakes in all datasets under all settings, and done so without sacrificing 1/3 of the dataset capacity; where the only other fix addresses 1 type of error in 1 dataset under 1 setting by removing 998 out of 3000 cases from the original dataset.
>
> And we sincerely hope our reviewer may agree with this analysis and provide us a more reflective rating, or let us know what are the remaining concerns. Thanks in advance.

---

### Decision · Program_Chairs · 2024-09-26

**Decision:**

Reject

**Comment:**

The primary aim of this paper is to highlight and rectify notable shortcomings in the MQuAKE dataset, which is commonly utilized for assessing multi-hop knowledge editing techniques in large language models. While the study effectively points out critical issues, proposes solutions, and presents benchmarking results, it also raises several concerns regarding its overall contribution, presentation, and comprehensive analysis.

Limited Novelty and Contribution:

Reviewers observed that although the paper tackles important topics, its contributions seem minimal, primarily revisiting the established MQuAKE dataset without introducing significantly innovative methods or insights. Key issues identified, such as the contamination between edited and unedited cases, have been previously discussed in the literature.
The authors referenced prior works that conducted less thorough error audits than this study; however, reviewers remained doubtful about the uniqueness and innovation of the proposed solutions and benchmarks.
Inadequate Discussion of Related Works:

Reviewers noted that the paper fails to sufficiently compare its methods with major related studies in the field, particularly those assessing multi-hop editing techniques, which undermines its credibility.
Additional works mentioned by reviewers, including RippleEdits, indicate that MQuAKE is not the sole dataset for such audits, raising concerns about the thoroughness of the existing methodologies review.
Lack of Insight Generation:

Reviewers expressed disappointment regarding the absence of new insights arising from the correction of identified errors, suggesting that the motivation for the research was insufficient. The authors were asked to clarify how their findings could inform future research in multi-hop knowledge editing beyond simply amending dataset errors.
While the authors attempted to introduce new metrics and benchmarks, reviewers felt a more defined pathway toward future developments and insights was warranted.
Presentation and Clarity Issues:

Significant concerns were raised about the presentation of results, particularly related to the clarity of experimental findings in Tables 3 and 4, which reviewers found challenging to interpret. Suggestions were made to enhance readability and structure, particularly regarding the excessive use of bold text and confusing table formats.
The absence of visual aids also hindered the understanding of results, and reviewers recommended incorporating figures to improve comprehension.
Limited Scope of Evaluation Benchmarks:

Several reviewers pointed out that the paper’s experimental evaluation, particularly concerning existing methods, was somewhat narrow. Despite demonstrating some improvements, the authors did not benchmark several emerging techniques due to their unavailability, which was viewed as a potential limitation in the evaluation approach.
Unfounded Claims About Data Integrity:

Questions arose regarding how authors guaranteed the accuracy of their "audited" dataset. Concerns were expressed about whether the verification was executed manually or automatically, and the criteria for determining "Conflicting Edits" were unclear. This lack of transparency contributed to doubts regarding the reliability of the corrections made.
While the significance of the issues addressed in the paper is acknowledged, the feedback indicates notable limitations in terms of novelty, rigor, and analytical depth that collectively weaken the case for acceptance.